# Cytosolic CRISPR RNAs for efficient application of RNA-targeting CRISPR-Cas systems

Ezra C K Cheng [ID], Joe K C Lam [ID] & S Chul Kwon [ID] [✉]

## Abstract

Clustered regularly interspaced short palindromic repeats/CRISPR-associated protein (CRISPR/Cas) technologies have evolved rapidly over the past decade with the continuous discovery of new Cas systems. In particular, RNA-targeting CRISPR-Cas13 proteins are promising single-effector systems to regulate target mRNAs without altering genomic DNA, yet the current Cas13 systems are restrained by suboptimal efficiencies. Here, we show that U1 promoter-driven CRISPR RNAs (crRNAs) increase the efficiency of various applications, including RNA knockdown and editing, without modifying the Cas13 protein effector. We confirm that U1-driven crRNAs are exported into the cytoplasm, while conventional U6 promoter-driven crRNAs are mostly confined to the nucleus. Furthermore, we reveal that the end positions of crRNAs expressed by the U1 promoter are consistent regardless of guide sequences and lengths. We also demonstrate that U1-driven crRNAs, but not U6-driven crRNAs, can efficiently repress the translation of target genes in combination with catalytically inactive Cas13 proteins. Finally, we show that U1-driven crRNAs can counteract the inhibitory effect of miRNAs. Our simple and effective engineering enables unprecedented cytosolic RNA-targeting applications.

**Keywords** CRISPR-Cas13; U1 Promoter; crRNA; Gene Expression Regulation; RNA Editing
**Subject Categories** Methods & Resources; RNA Biology; Translation & Protein Quality

## Introduction

There have been known two RNA-targeting CRISPR-Cas systems, which are the Type III and VI systems (Makarova et al, 2020). Yet, only the Type VI Cas13 systems display exclusive RNase activity as the Type III Csm systems carry an additional DNase domain (Samai et al, 2015). The Cas13 family is classified as a Class 2 system for their single effector protein (Makarova et al, 2020). The RNase function of the Cas13 protein is activated by a conformational change of the two HEPN (Higher Eukaryotes and Prokaryotes Nucleotide-binding) domains upon target RNA binding (East-Seletsky et al, 2016; O'Connell, 2019; Pillon et al, 2021). The unique functionality of Cas13 systems has led to the development of numerous novel technologies that DNA-targeting systems, such as Cas9, could not provide. The Cas13 systems have been widely used for rapid detection of viral RNAs (Ackerman et al, 2020; Freije et al, 2019; Gootenberg et al, 2017; Patchsung et al, 2020), precise in vivo RNA knockdown (Kushawah et al, 2020), and effective degradation of repeat RNA aggregates in repeat expansion disorders (Morelli et al, 2023; Zhang et al, 2020). The use of catalytically dead Cas13 (dCas13) proteins as a target-specific RNA-binding module has also enabled new methods, including RNA editing (Abudayyeh et al, 2019; Cox et al, 2017; Kannan et al, 2022; Nakagawa et al, 2022; Xu et al, 2021), epitranscriptomic regulation (Rauch et al, 2018; Roundtree et al, 2017; Wilson et al, 2020), and alternative splicing modulation (Fiflis et al, 2024; Konermann et al, 2018). Recently, the Iwasaki group showed that precise translational repression without severe off-target effects could also be achieved with dCas13 proteins (Apostolopoulos et al, 2024).

In humans, the expression of the Sm-class U1 small nuclear RNA (snRNA) is mediated by RNA polymerase II (RNAPII) (Will and Lührmann, 2011). The U1 snRNA promoter does not contain any TATA-box, but the distal sequence element (DSE) and the proximal sequence element (PSE) have been identified within the U1 promoter for efficient transcription of U1 pre-snRNA (Hernandez, 2001). The 3′ end processing is mediated by the 3′ box, which is a short motif generally between 13 to 16 nucleotides and located downstream of the U1 snRNA sequence (de Vegvar et al, 1986; Hernandez, 1985; Uguen and Murphy, 2003, 2004). In contrast to the polyadenylation signal-dependent 3′ end processing of CMV or EF1α RNAPII-transcripts, the 3′ box-dependent processing pathway utilizes a distinct cleavage complex called Integrator, enabling efficient expression of short RNA species without a long poly(A) tail (Chen and Wagner, 2010; Matera and Wang, 2014). Following transcription in the nucleus by RNA polymerase II, the 5′ end of a U1 pre-snRNA is chemically modified with a 7-methylguanosine (m⁷G) cap structure. Like the regular RNA polymerase II-transcribed mRNAs, the m⁷G cap then mediates the export of the snRNA into the cytoplasm through the nuclear pore complex by signaling the assembly of a multi-protein export complex containing CBC (cap-binding complex), ARS2 (arsenite resistance protein 2), PHAX (phosphorylated adapter RNA export), CRM1 (chromosome region maintenance 1), and an active Ran-GTPase (Ras-like nuclear guanosine triphosphatase) (Ohno et al, 2000). Upon successful nuclear export, the export complex dissociates from the 5′ end of the U1 snRNA, and then the heptameric Sm core proteins and the SMN (survival motor

School of Biomedical Sciences, LKS Faculty of Medicine, The University of Hong Kong, Hong Kong SAR, China. ✉E-mail: chul@hku.hk

neuron) complex assemble around the Sm site (Pellizzoni et al, 2002). After that, TGS1 (trimethylguanosine synthase 1) is recruited to the snRNA for hypermethylation of the m⁷G cap to make the trimethylguanosine (TMG; $m^{2,2,7}G$) cap which signals the nuclear import by recruiting SPN1 (Snurportin-1) and Importin-β (Huber et al, 1998; Mouaikel et al, 2002). As a result, the U1 snRNA is transported back into the nucleus and localized at Cajal bodies for the final maturation steps before proceeding to spliceosome assembly (Staněk and Neugebauer, 2006).

Until now, the most widely used expression system for CRISPR-Cas guide RNAs in human cells is the RNA polymerase III-dependent U6 promoter. It has been successfully repurposed to express short hairpin RNAs (shRNAs) (Paddison et al, 2002), single-guide RNAs (sgRNAs) of Cas9 (Ran et al, 2013), and CRISPR RNAs (crRNAs) of Cas12 (Zetsche et al, 2015). Some groups have reported the use of alternative promoters for Cas9 guide RNA. For example, Lu and colleagues reported the adaptation of the CMV promoter for conditional expression of Cas9 guide RNAs in 2014 (Nissim et al, 2014). Rinn and colleagues also tested CMV, EF1α, and U1 promoters to express unconventionally long guide RNAs in 2015 (Shechner et al, 2015). Especially for the U1 promoter, the authors included the Sm-binding site and the downstream stem-loop so that the guide RNA can be re-imported into the nucleus after processing (Shechner et al, 2015). The canonical mRNA promoters, such as CMV and EF1α promoters, also have been shown to be useful for multiplexed crRNA expression for Cas12a, which contains the pre-crRNA processing activity; however, it is still controversial whether these RNAPII promoters outperform the U6 promoter for single gene regulation (Campa et al, 2019; Zhao et al, 2024; Zhong et al, 2017). The first RNA-targeting Cas13 system that is functional in human cells was reported in 2017 (Abudayyeh et al, 2017), and the authors tested U6 and tRNA promoters for crRNA expression. Afterward, the U6 promoter has been widely adopted for various Cas13 systems (Abudayyeh et al, 2019; Ackerman et al, 2020; Cox et al, 2017; Freije et al, 2019; Wilson et al, 2020).

In contrast to the U1 snRNA, the endogenous U6 snRNA is thought to remain in the nucleus throughout its biogenesis pathway (Pessa et al, 2008; Vankan et al, 1990). Therefore, it remains unknown whether the conventional U6 promoter is the most effective method to express crRNAs for cytosolic applications of RNA-targeting Cas systems. Interestingly, a recent study showed that nucleocytoplasmic shuttling Cas13d could increase the cytosolic viral RNA degradation efficiency of U6-driven crRNAs (Gruber et al, 2024). Here, we demonstrate that a well-designed U1 promoter is able to express cytosolic crRNAs, thereby boosting the efficiency of various cytosolic Cas13 applications in a simple and robust manner.

## Results

### Repurposing U1 promoter for cytosolic crRNA

To enable the cytosolic distribution of Cas13 crRNA, we adopted the U1 snRNA promoter and associated 3′ elements, which had been utilized to express short hairpin RNAs (shRNAs) in 2004 (Denti et al, 2004). Our design includes the human U1 promoter (~400 bp), +1 adenosine for preferential transcription initiation, a

Cas13 crRNA sequence, the U1 3′ downstream region, and the U1 3′ box (Figs. 1A and EV1A). We hypothesized that the removal of the Sm site prevents assemblies of the Sm core and the SMN complex, thus circumventing the nuclear import machinery and enabling cytosolic localization of Cas13 crRNA (Fig. 1B). First, we used the CasRx effector, which is also known as RfxCas13d (Cas13d from *Ruminococcus flavefaciens* XPD3002), as a model system. To confirm the localization pattern of the CasRx protein, we compared CasRx tagged with the SV40 nuclear localization signal (NLS) at both N- and C-termini and CasRx without any localization signal, which are termed CasRx-NLS and CasRx, respectively. To evaluate their localization independently of each other, we transiently transfected either the CasRx protein only or crRNA only (Fig. EV2A). We used a precursor crRNA (pre-crRNA) containing 4 copies of the same crRNA sequence to amplify the RNA FISH signal. The localization of CasRx-NLS was found to be restricted within the nucleus, while CasRx was distributed primarily in the cytoplasmic fraction (Fig. EV2A, top). The U6 promoter-driven crRNA showed strong signals in the nucleus in the absence of the protein partner. However, the U1 promoter-driven crRNA showed much weaker signals overall, and there was no clear evidence of the cytosolic localization pattern (Fig. EV2A, middle and bottom; Fig. EV2B).

We hypothesized that the U1 promoter-driven crRNA was rapidly degraded in the cytosol in the absence of the protein partner. To test this idea, we co-transfected the pre-crRNA and CasRx proteins and observed the localization of crRNA. Interestingly, the cytosolic abundance of U1-driven crRNA was dramatically increased by CasRx without NLS (Fig. EV2C). In contrast, U6-driven crRNA was still preferentially observed in the nucleus in the presence of CasRx without NLS (Fig. EV2C). This strongly suggests that U1-driven crRNAs can be exported to the cytoplasm and stabilized by the CasRx effector protein. The idea of crRNA stabilization through the effector protein is further confirmed by concurrent staining of crRNA and the CasRx effector protein in the same cells (Figs. 1C and EV3A). Co-localization of U1-driven crRNA and CasRx without NLS was clearly observed in the cytosol (Fig. 1C, bottom); however, most U6-driven crRNA was localized in the nucleus in the presence of CasRx without NLS (Fig. 1C, top). Taken together, our results suggest that the U1 promoter without the Sm site can express crRNA that is effectively exported to the cytosol.

To confirm the protein-mediated crRNA stabilization event, we carried out northern blotting after transfection of the mature form of crRNAs and CasRx proteins. As expected, the existence of protein partners increased the amount of crRNA because naked RNAs are prone to degradation in cells (Figs. 1D,E and EV3B). Interestingly, we found that the U6-driven crRNA was more stabilized by CasRx-NLS than CasRx, and the U1-driven crRNA was better protected by CasRx than CasRx-NLS (Fig. 1D,E), suggesting that the major subcellular locations of U6 crRNA and U1 crRNA are the nucleus and the cytosol, respectively. We also found that U6-driven crRNAs showed a single homogeneous length distribution (Fig. 1D, lanes 2–4). In contrast, U1-driven crRNAs had a heterogeneous length distribution in the presence of CasRx-NLS (Fig. 1D, lane 6). The smaller RNA fragment of the U1-driven crRNA was strongly stabilized by CasRx without NLS (Fig. 1D, lane 7), suggesting that cytosolic CasRx effectively binds to the U1-driven crRNA in the cytoplasm, and the uncovered part of crRNA

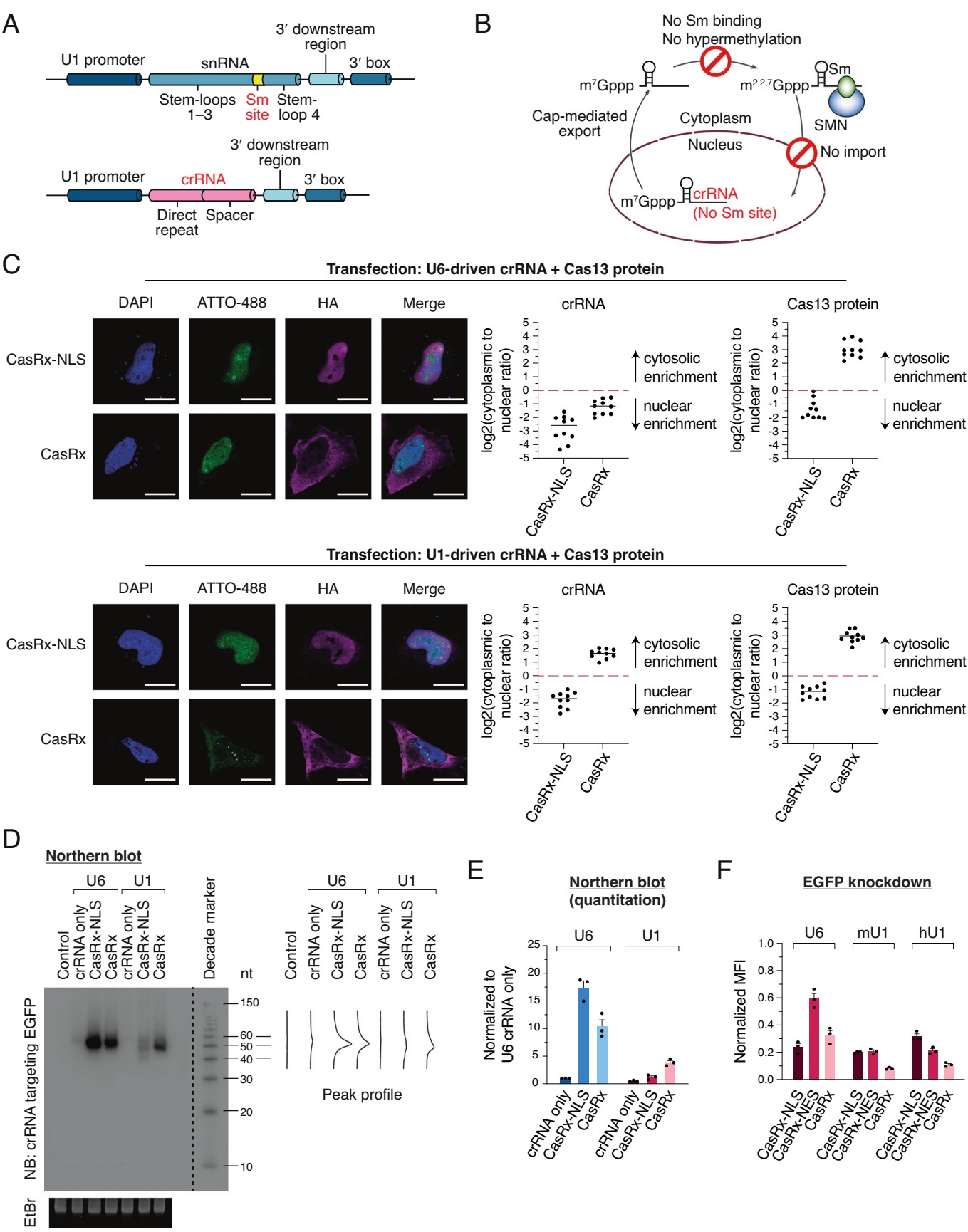

**Figure 1.   Repurposing U1 promoter for cytosolic crRNA.**

(A) Schematic of repurposing a U1 promoter for crRNA transcription. (B) Schematic of the cytosolic enrichment of crRNA from the repurposed U1 promoter. (C) Subcellular localization of crRNA targeting mCherry in the presence of the CasRx protein. RNA FISH was performed following immunocytochemistry of the same cells. Note that mCherry was not expressed and acted as a non-human sequence. Nuclei were stained with DAPI. crRNA targeting mCherry was visualized with an ATTO-488-conjugated probe. CasRx effector was stained with anti-HA antibody. Total fluorescence intensity is shown as a scatter plot. Mean is indicated as a horizontal line. $n = 10$ from three biological replicates. Scale bars: 20 μm. (D) Northern blot showing crRNA expression and length distribution. crRNA targeting EGFP was visualized with a $^{32}$P-labeled probe. Note that EGFP was not expressed and acted as a non-human sequence. (E) Northern blot quantitation. Mean and s.e.m. are shown ($n = 3$, biological triplicates). (F) EGFP knockdown by combinations of CasRx effectors, crRNAs, and promoters. Median fluorescence intensity (MFI) of each combination is normalized to the corresponding CasRx effector in combination with non-targeting crRNA. mU1, mouse U1; hU1, human U1. Mean and s.e.m. are shown ($n = 3$, biological triplicates). Source data are available online for this figure.

is trimmed by cellular RNases, such as DIS3L2 (Łabno et al, 2016; Pirouz et al, 2016; Ustianenko et al, 2016). The homogeneous length distribution of the U6-driven crRNA may be explained by the 3′ end protection with the La protein in the nucleus (Yan et al, 2024).

Next, we tested whether the co-localized crRNA and CasRx effector proteins can improve the RNA knockdown efficiency. The highest EGFP knockdown efficiency of the U6-driven crRNA was achieved with CasRx-NLS, but that of the U1-driven crRNA was obtained with CasRx, suggesting the importance of the CasRx–crRNA co-localization (Fig. 1F). Interestingly, the cytosolic pair of U1-driven crRNA and CasRx showed a higher level of EGFP knockdown than the nuclear counterpart of U6-driven crRNA and CasRx-NLS. The mouse U1 promoter-driven crRNA showed the same pattern as the human U1-driven crRNA, indicating that the U1 promoter engineering principle is robust (Figs. 1F and EV1B). We also included a C-terminal NES (nuclear export signal)-fused CasRx construct and obtained the same pattern, albeit its overall knockdown efficiency was lower than that of CasRx without NLS, suggesting that the hydrophobic NES residues may affect the protein folding or the RNA binding (Fig. 1F). Taken together, our data demonstrate that U1-driven crRNAs are preferentially localized in the cytosol and can be efficiently coupled with cytosolic CasRx proteins.

## End sites of U1-driven crRNA

Based on the relative position of the direct repeat hairpin and the variable spacer sequence, Cas13 crRNAs can be classified into two types. Cas13a and Cas13d families have the direct repeat hairpin at the 5′ end, and the Cas13b family has the hairpin at the 3′ end. To test whether the repurposed U1 promoter can produce various crRNAs with different spacer sequences and lengths in a consistent manner, we carried out rapid amplification of cDNA ends (RACE) experiments (Fig. 2A). HEK293E cells were transiently transfected only with crRNA constructs to exclude the secondary effects of Cas13 proteins, which can cleave or stabilize some part of crRNAs (Fig. 1D,E) (East-Seletsky et al, 2016; Slaymaker et al, 2019).

First, we conducted 3′ RACE to determine the 3′ end position of various U1-driven crRNAs. Many studies have examined the 3′ processing of the endogenous human U1 snRNA (Uguen and Murphy, 2003, 2004); however, it remains unclear whether the U1 snRNA-coding part is completely dispensable for the precise 3′ end processing. We used a U6-driven CasRx crRNA targeting the EGFP sequence as a positive control. As expected, the 3′ end of this RNA

polymerase III-driven crRNA was located after the poly(U) sequences (Figs. 2B and EV4A) (Bogenhagen and Brown, 1981). In contrast, for all U1-driven crRNAs tested here, the most prevalent 3′ end sequences contained template-derived downstream 3 nucleotides ("ACT") at the 3′ end of the crRNA sequence, regardless of different locations of the direct repeat ("CasRx", "PspCas13b", or "Cas13bt1"), spacer sequences (targeting "EGFP" or "KRAS"), spacer lengths ("extended"), or stem-loop structures ("SL4") (Fig. 2A,B) (Cox et al, 2017; Kannan et al, 2022). The 3′ end position was located at 7 nt upstream from the 3′ box (Figs. 3B and EV1A), and it was the same site where the endogenous U1 pre-snRNA is cleaved in vivo (Hernandez, 1985). Taken together, we conclude that neither the replacement of the U1 snRNA sequence nor the removal of the Sm site affects the preferred 3′ end site, as long as the 3′ downstream region and the 3′ box are preserved together with the U1 promoter.

While analyzing the 3′ RACE results, we frequently observed non-templated thymidines at the 3′ end of U1-driven crRNAs (Figs. 2C and EV4B), suggesting that they are uridylation substrates of terminal uridylyltransferase TUT4 and TUT7, which are preferentially localized in the cytosol (Lim et al, 2014). The uridylation ratio was high on the CasRx crRNAs, which contain single-stranded spacer sequences at the 3′ end (Fig. 2C, crRNAs 2–5). In contrast, it was relatively low on PspCas13b and Cas13bt1 crRNAs, having the direct repeat hairpin at the 3′ end (Fig. 2C, crRNAs 6–9). Non-modified crRNAs and mono/di-uridylated crRNAs comprised over 80% of the total population (Fig. 2C), indicating that the exact 3′ end sequence of a U1-driven crRNA is predictable.

Next, we carried out 5′ RACE to find the 5′ end position of crRNAs driven by the U1 promoter (Fig. 3A). Non-templated three guanosines upstream of the first nucleotide in the crRNA were expected due to the non-templated addition of three cytosines at the 3′ end of cDNA by the template-switching activity of reverse transcriptases (Wulf et al, 2019). To validate the experimental system, we used a U6-driven Cas13bt1 EGFP crRNA (Fig. 3B, crRNA 14). As expected, the U6-driven crRNA showed a highly homogeneous population of the 5′ end sequences at +1 G (Fig. 3B) and non-templated three G sequences at the 5′ end (Fig. 3C). For the U1-driven crRNAs, to our surprise, four non-templated G sequences were found as much as three G sequences at the 5′ end of Cas13bt1 and PspCas13b crRNAs (Figs. 3C and EV4D). It has been known that 5′ capped RNAs make four-guanosine-containing cDNAs after template-switching reverse transcription (Wulf et al, 2019), implying that half of the U1-driven crRNAs in cells are 5′ capped.

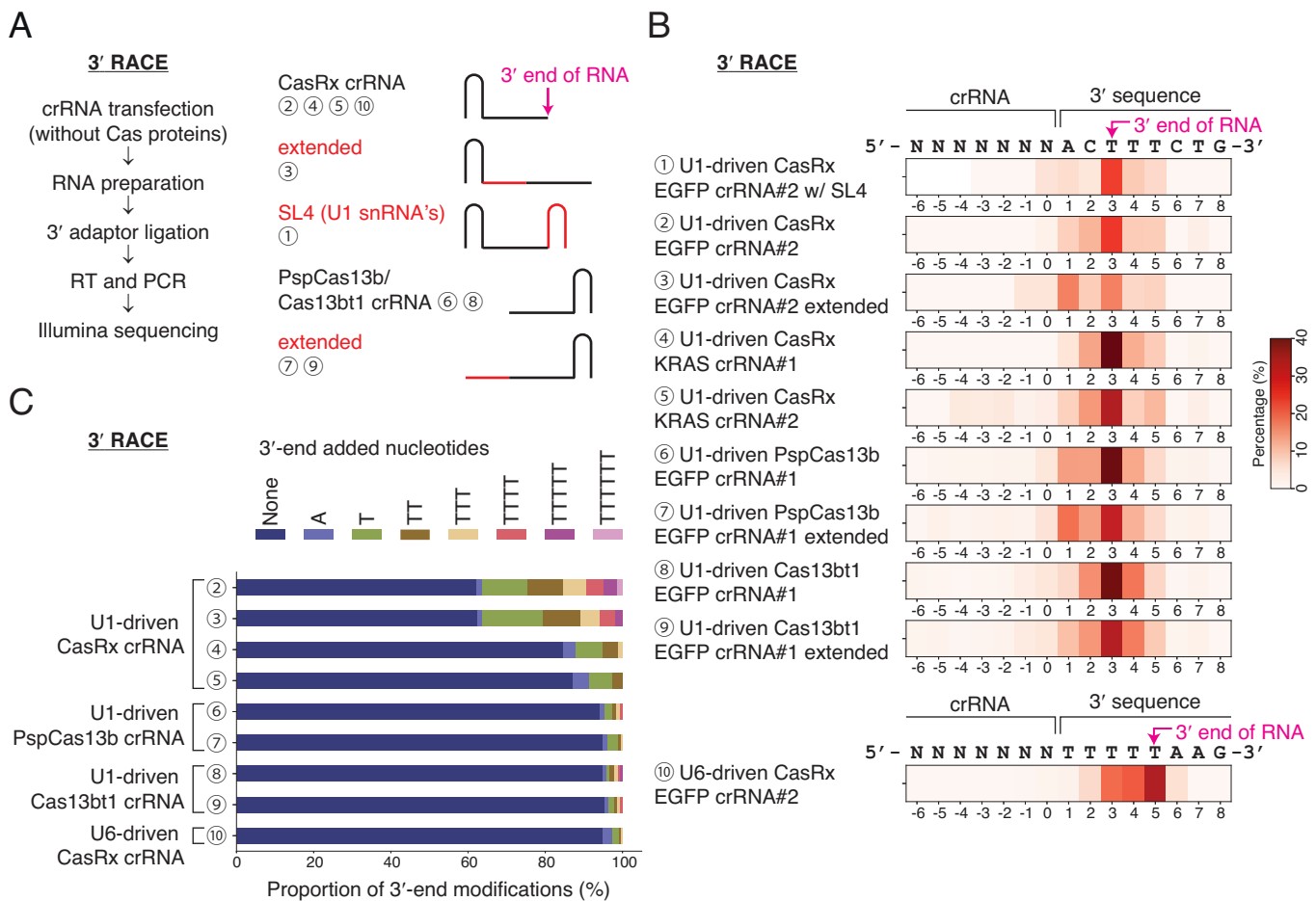

**Figure 2. 3′ end sites of U1-driven crRNA.**

(A) Schematic of 3′ RACE. Circled numbers indicate crRNAs in B. SL4, the stem-loop 4 hairpin of the U1 snRNA. (B) Analysis of 3′ end positions. The proportion of each 3′ end position is shown as a heatmap. (C) Analysis of 3′ end modifications.

To validate the presence of m⁷G cap on U1-driven crRNA, we performed an in vitro pull-down assay to capture 5′ capped crRNA using the recombinant mouse $eIF4E_{K119A}$ protein (Figs. 3D and EV4E,F) (Trotman et al, 2017). The K119A mutation has been shown to increase the cap binding affinity (Choi and Hagedorn, 2003). U1-driven Cas13bt1 crRNA showed a clear enrichment in the bound fraction as the m⁷G capped GAPDH mRNA, while the pattern for U6-driven crRNA resembled the non-capped tRNA Leu-AAG and endogenous U6 snRNA transcripts (Fig. 3D). Similar enrichment pattern of U1-driven crRNA was reproduced using CasRx crRNA, demonstrating that the 5′ capping of U1-driven crRNA is not affected by the 5′ direct repeat hairpin structure (Fig. EV4E).

We also tested whether the +1 position nucleotide affects the decision of the transcription start site. The +1 G Cas13bt1 crRNA provided a homogeneous 5′ end population as much as the +1 A crRNA (Fig. 3B, crRNAs 8 and 11); however, the +1 C and +1 T Cas13bt1 crRNAs showed heterogeneous 5′ end sequences, and the major transcription start site was shifted to one nucleotide upstream (Fig. 3B, crRNAs 12 and 13). In addition, the crRNA spacer length did not affect the homogeneity of the 5′ end crRNA

positions (Figs. 3B and EV4C, crRNAs 7 and 9). These data indicate that homogeneous 5′ capped crRNAs can be produced using the U1 promoter with +1 A or +1 G sequences.

Given the RACE analysis results, we hypothesized that the target knockdown activity of crRNA is correlated with the homogeneity of 5′ end sequences, which may affect the expression level. A positive correlation between the 5′ end sequence homogeneity and the EGFP knockdown efficiency was shown with the U1-driven Cas13bt1 EGFP crRNAs carrying alternative +1 nucleotides (Fig. 3E). Moreover, the +1A crRNA had a better knockdown effect than the +1 G crRNA, suggesting that the +1 A transcription start site is ideal for U1-driven crRNAs (Fig. 3E).

The optimum U1-driven crRNA spacer length was determined by the same EGFP knockdown experiments. For Cas13bt1 and PspCas13b, 30-nt spacer crRNAs showed better knockdown efficiency than 50-nt crRNAs (Fig. 3E), possibly because the extended sequence folded back onto the crRNA and inhibited the binding of target RNA. A similar result was obtained with CasRx crRNAs, where a 23-nt crRNA performed better than a 50-nt crRNA (Fig. EV4G). The optimum U1-driven crRNA spacer lengths were consistent with the previous data from U6-driven

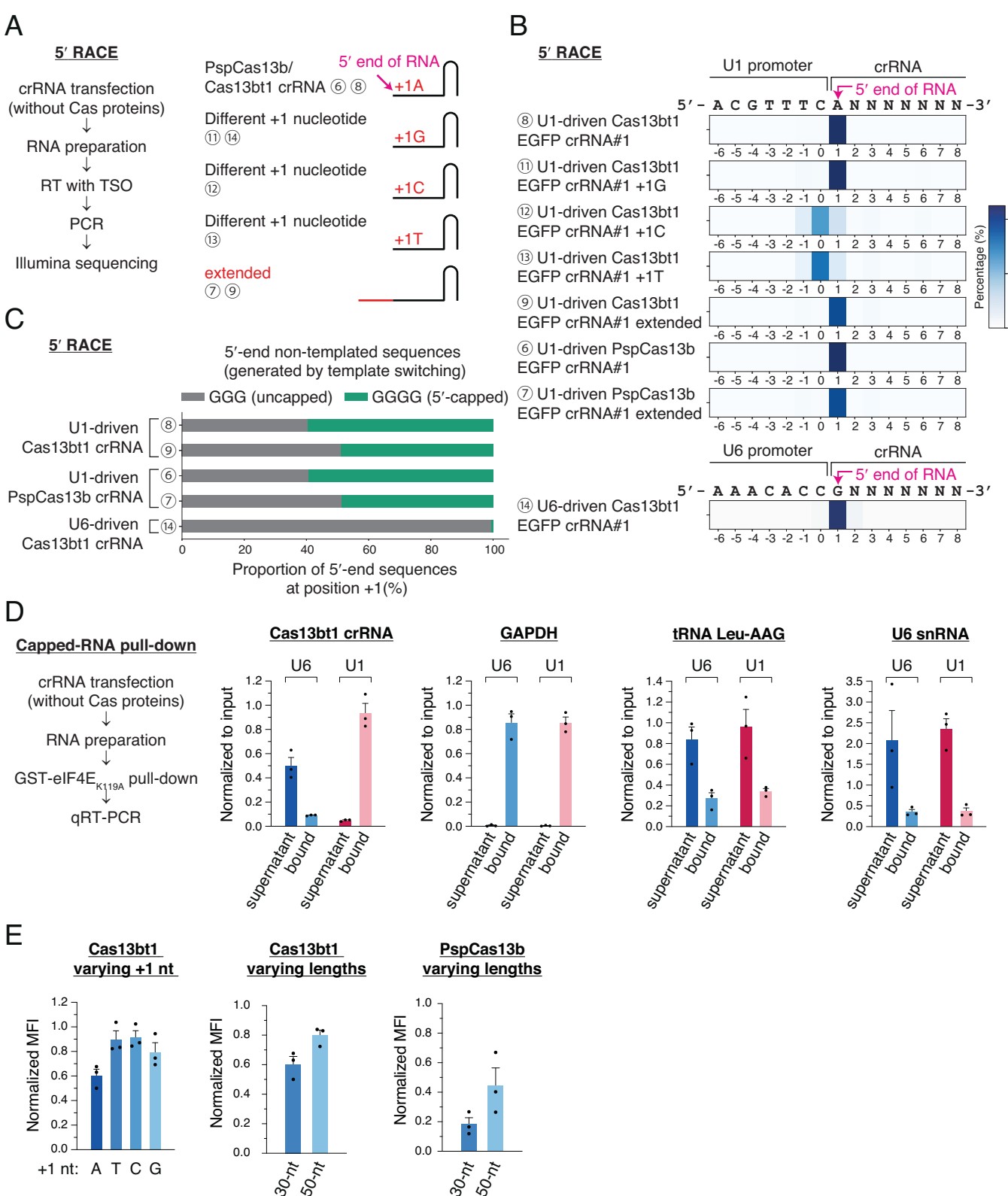

Figure 3.  5′ end sites of U1-driven crRNA.

(A) Schematic of 5′ RACE. Circled numbers indicate crRNAs in B. TSO, template-switching oligo. (B) Analysis of 5′ end positions. The proportion of each 5′ end position is shown as a heatmap. (C) Analysis of 5′ end non-templated sequences. (D) qRT-PCR following pull-down assay of capped RNA with recombinant mouse eIF4E$_{K119A}$ protein. Enrichment of U1-driven Cas13bt1 crRNA in the bound fraction is compared with U6-driven Cas13bt1 crRNA. GAPDH mRNA is included to represent capped RNA, while tRNA Leu-AGG and endogenous U6 snRNA are included to represent non-capped RNA. Mean and s.e.m. are shown ($n = 3$, biological triplicates). (E) EGFP knockdown by combinations of Cas13 effectors and U1-driven crRNAs. Median fluorescence intensity (MFI) is shown. Experiments for the left and middle plots were done simultaneously, and the +1 A 30-nt crRNA sample served as a control for both plots. Mean and s.e.m. are shown ($n = 3$, biological triplicates). Source data are available online for this figure.

crRNAs (Wessels et al, 2020), indicating that the identical Cas13 crRNA spacer sequences can be used for both U1 and U6 promoters.

## Enhancing RNA knockdown and editing with cytosolic crRNAs

Given the ability of U1-driven crRNA to improve the EGFP knockdown efficiency with co-localized CasRx effector protein (Fig. 1F), we sought to expand the scope of utility to different Cas13 effectors and applications. Many existing Cas13 technologies have been using the NES-tagged versions of effector proteins since the initial report of Cas13 for RNA-targeting application (Cox et al, 2017). Especially, PspCas13b, PguCas13b, and RanCas13b showed the best performance with an NES tag rather than an NLS tag in combination with U6-driven crRNAs (Cox et al, 2017).

The dual-luciferase reporter assay results showed that U1-driven crRNAs targeting *Gaussia* luciferase (Gluc) were able to exhibit higher knockdown activity with all of the three cytosolic Cas13 effectors tested here, including PspCas13b, Cas13bt1, and RanCas13b (Fig. 4A). Next, we tested the knockdown activity on endogenous mRNAs using qRT-PCR. U1-driven crRNAs targeting KRAS mRNA were again able to produce a higher knockdown efficiency with all of PspCas13b, Cas13bt1, and RanCas13b (Fig. 4B).

RNA editing with CRISPR-Cas13 is another fascinating technology for biology and therapy. Given that most of the effective editing effectors, such as dPspCas13b and dRanCas13b, have been tagged with NES (Cox et al, 2017), we hypothesized that the cytosolic U1-driven crRNA is able to increase the editing efficiency. Interestingly, an approximately 3-fold boost in editing efficiency was achieved with U1-driven crRNAs for both A-to-I and C-to-U RNA editing systems using three different RNA editors (Fig. 4C), indicating that the new crRNA expression system is robust and reliable.

Furthermore, as mentioned above in the RACE analysis, the transcription termination of the U6 promoter is signaled by a poly(U) stretch. Therefore, any short stretch of four repeating uridines in the crRNA sequence causes the early termination of U6-driven crRNA transcription, making it less efficient than its maximum potential. This problem was initially raised for *Streptococcus pyogenes* Cas9 (SpCas9), whose direct repeat contains four consecutive U sequences, and later improved by using alternative nucleotides (Chen et al, 2013; Hsu et al, 2013). We speculated that such repeating uridines are tolerated in the transcription of U1-driven crRNA because the transcription termination is mediated by the downstream 3′ regulatory sequences. We looked through the NCBI ClinVar database for pathogenic single nucleotide variants (SNV) that can be targeted by the existing REPAIR (A-to-I editing) or RESCUE (C-to-U editing)

RNA-editing tools (Fig. EV5A,B) (Abudayyeh et al, 2019; Cox et al, 2017). Intriguingly, we found that out of the 13,238 targetable pathogenic SNVs, 17.0% of the point mutations were located adjacent to an AAAA sequence, meaning their correction is likely less efficient with the U6-driven crRNA. Taken together, our U1-driven crRNA can confer the maximum RNA editing efficiency without having restrictions on target site selection.

## Advancing cytosol-specific RNA modulations

During the life cycle of an mRNA, several events, such as translation, occur exclusively in the cytoplasm. A recent paper clearly demonstrated that precise translational repression can be achieved with catalytically dead Cas13 proteins targeting the start codon (AUG) (Apostolopoulos et al, 2024). The method can exclude possible off-target effects from activated Cas13 HEPN domains, which cleave any nearby RNAs by so-called "collateral cleavage activity" (East-Seletsky et al, 2016; Slaymaker et al, 2019; Zhang et al, 2018). We hypothesized that U1-driven crRNA can boost the translational repression effect by dead Cas13. To prove this, we first confirmed the functionality of Cas13 crRNAs targeting the AUG site, the coding sequence (CDS), and the 3′ untranslated region (3′ UTR) in terms of RNA knockdown using catalytically active PspCas13b (Fig. 5A, left). As expected, U1-driven crRNAs showed better knockdown efficiency than U6-driven crRNAs (Fig. 5A, left). Next, we tested the translational inhibition activity using dead PspCas13b. In our experimental condition, U6-driven crRNAs could not block the translation, but U1-driven crRNAs targeting the start codon efficiently decreased the expression of EGFP (Fig. 5A, right). We repeated the experiment using *Gaussia* luciferase as another target gene and obtained a similar result (Fig. EV6A). More specifically, both U6 and U1-driven crRNAs could effectively reduce the translation, but U1-crRNAs outperformed U6-crRNAs (Fig. EV6A). We also tested several crRNAs targeting the 5′ UTR of EGFP mRNA and confirmed that the start codon is the best target site for translational repression (Fig. EV6B), which is consistent with the recent study (Apostolopoulos et al, 2024).

As a second proof of concept, we aimed to intervene in the microRNA (miRNA)–target interactions, which take place mainly in the cytoplasm (Zeng and Cullen, 2002). We hypothesized that the competitive binding of dPspCas13b at miRNA-binding sites can prevent the degradation of the target mRNA, thereby restoring the target expression (Fig. 5B). To test this idea, we used a firefly luciferase reporter containing the binding sites of miR-1 or miR-302a, which are known to be not expressed in HEK293E cells. As expected, the U1-driven crRNAs achieved a greater extent of expression restoration than the U6 counterparts for both miR-1 and miR-302a reporters (Fig. 5B), highlighting the possibility of regulating specific miRNA-binding sites in the cytoplasm.

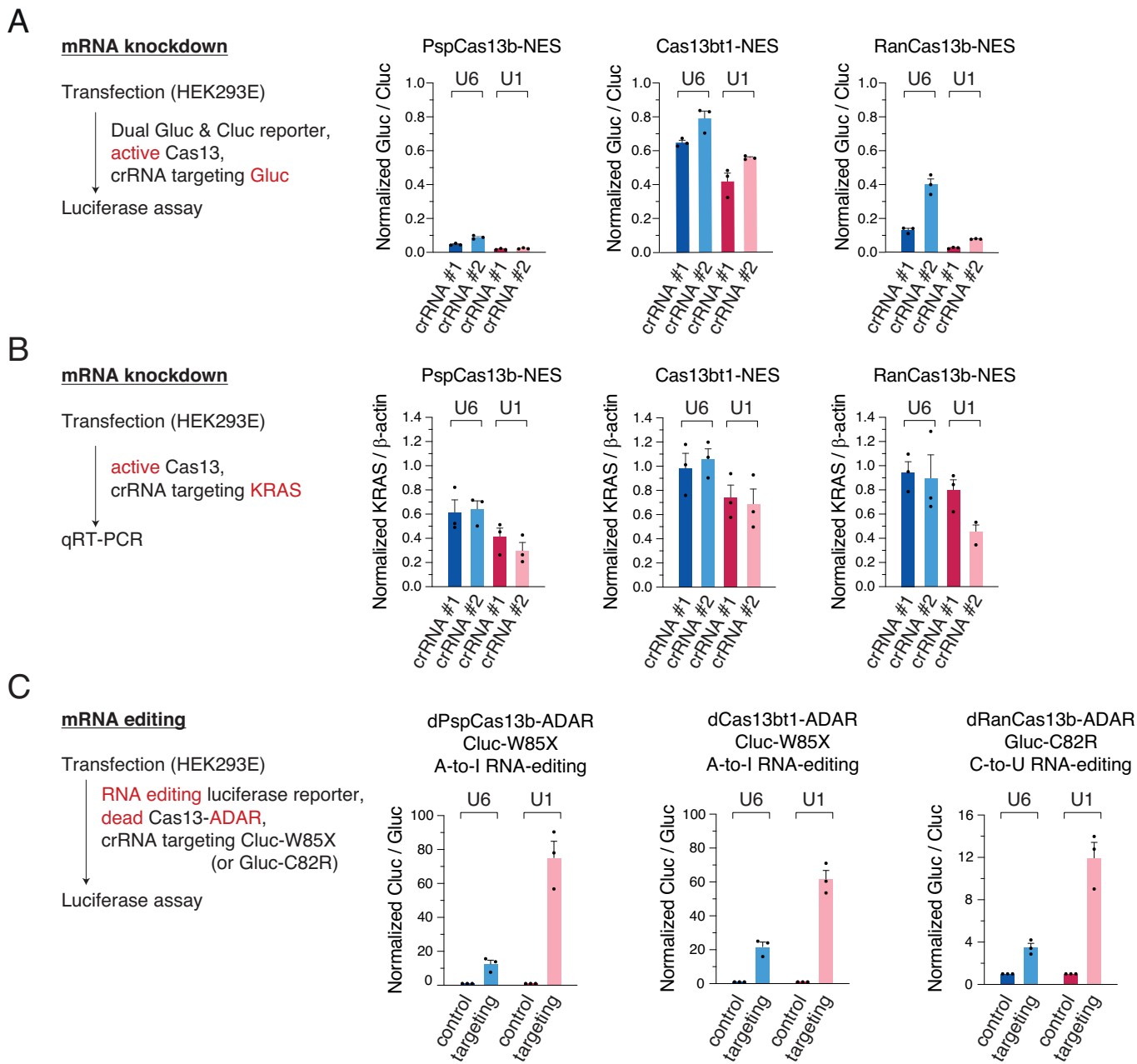

**Figure 4. Enhancing RNA knockdown and editing with cytosolic crRNAs.**

(A) Dual-luciferase reporter assay with U1-driven Gluc crRNAs for PspCas13b, Cas13bt1, and RanCas13b. Knockdown efficiency with U1-driven crRNA is compared with the efficiency with U6-driven crRNA. Different mRNA loci are targeted by crRNA #1 and crRNA #2. Cluc, *Cypridina* luciferase. Gluc, *Gaussia* luciferase. Mean and s.e.m. are shown ($n = 3$, biological triplicates). (B) qRT-PCR with U1-driven KRAS crRNAs for PspCas13b, Cas13bt1, and RanCas13b. Knockdown efficiency with U1-driven crRNA is compared with the efficiency with U6-driven crRNA. Different mRNA loci are targeted by crRNA #1 and crRNA #2. Mean and s.e.m. are shown ($n = 3$, biological triplicates). (C) Dual-luciferase reporter assay for A-to-I RNA editing using dPspCas13b-ADAR and dCas13bt1-ADAR, and C-to-U RNA editing using dRanCas13b-ADAR. Cluc, *Cypridina* luciferase. Gluc, *Gaussia* luciferase. Mean and s.e.m. are shown ($n = 3$, biological triplicates). Source data are available online for this figure.

## Discussion

In this study, we repurpose the U1 promoter for the expression of cytosolic Cas13 crRNAs and validate the advantage of U1-driven crRNAs over the conventional U6-driven crRNAs on existing RNA-targeting CRISPR-Cas13 technologies. We especially demonstrate that cytosol-specific gene regulation events, such as translation and miRNA-mediated repression, can be dramatically modulated by U1-driven crRNAs. Also, we identify the end sequences of U1-driven crRNAs for the reliable use of the new system by broad research communities.

Although the U1 promoter had been repurposed to express some exogenous RNAs (Denti et al, 2004; Shechner et al, 2015), it has not been widely utilized. In 2004, Bozzoni and colleagues

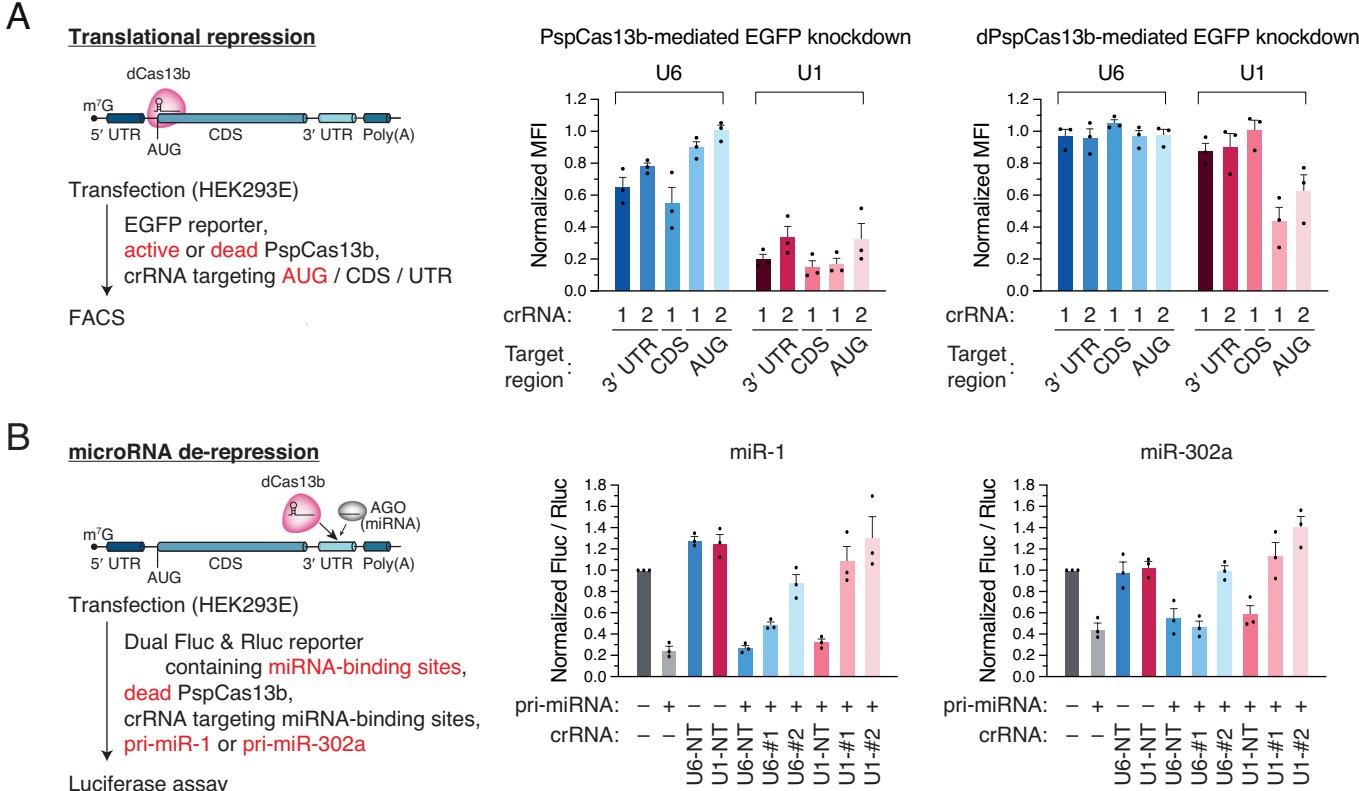

**Figure 5. Advancing cytosol-specific RNA modulations.**

(A) Translational repression using AUG-targeting crRNA and catalytically dead PspCas13b. Left, schematic. Right, flow cytometry of EGFP intensity. Repression efficiency with U1-driven crRNA is compared with the efficiency with U6-driven crRNA. Median fluorescence intensity (MFI) is normalized to the non-targeting control. Mean and s.e.m. are shown ($n = 3$, biological triplicates). (B) miRNA de-repression using crRNA targeting the miRNA-binding site and catalytically dead PspCas13b. Left, schematic. Right, dual-luciferase reporter assay using dPspCas13b. Fluc, firefly luciferase. Rluc, *Renilla* luciferase. NT, non-targeting. Mean and s.e.m. are shown ($n = 3$, biological triplicates). Source data are available online for this figure.

expressed shRNAs using the U1 promoter (Denti et al, 2004). The authors removed the Sm-binding site and the downstream stem-loop 4 but kept the promoter and the 3′ box element. Because the authors showed neither the shRNA localization nor the 5′ capping status, it has been unclear whether the U1 promoter-driven shRNA, which does not contain the endogenous U1 snRNA-coding sequence, is actively exported into the cytoplasm. Even if the shRNA is enriched in the cytoplasm, it can be interpreted by the action of the Exportin-5 pathway, which recognizes the double-stranded RNA (dsRNA) with 3′ overhang nucleotides. Most importantly, this U1-driven shRNA design has not been commonly used for the last two decades, possibly because the 5′ cap structure interferes with the recognition of DICER (Lee et al, 2023) and lowers the processing efficiency. The lack of knowledge about the exact end sequences driven by this engineered promoter has also made the application of this design to different guide sequences challenging. We are the first to adopt the design of the U1 promoter without the Sm site and stem-loop 4 for Cas13 crRNAs and provide sufficient evidence for its cytosolic localization, 5′ capping status, and, most importantly, superior ability (Fig. 6A–C).

The 3′ RACE analysis of U1-driven crRNA in the absence of the Cas13 effector protein shows the existence of the extra "ACT" sequences derived from the 3′ downstream region of the U1

promoter at the 3′ end of crRNA (Figs. 2B and 6A). However, the addition of these extra sequences does not seem to affect the function of crRNA in cells. For the Cas13b family members, the pre-crRNA processing activity of the Cas13b protein will cleave the crRNA after direct repeat, releasing the extra downstream nucleotides (Slaymaker et al, 2019). For U1-driven CasRx crRNA, the 3′ extra sequences seem to be trimmed in the cytosol by exonucleases, as shown in the northern blot (Fig. 1D, lanes 6 and 7).

Recent advances in CRISPR-Cas systems have opened up new possibilities for translational regulation tools. For example, Yeo and colleagues tethered UBAP2L (ubiquitin-associated protein 2-like) to RNA-targeting Cas9 (RCas9) and demonstrated the capability of promoting target protein translation through the interaction between UBAP2L and translating ribosomes (Luo et al, 2020). In addition, Liu and colleagues showed that linking the Cas13 crRNA to the SINEB2 (short interspersed element B2) domain of Uchl1 (ubiquitin C-terminal hydrolase L1) long noncoding RNA (lncRNA) could selectively promote translation of endogenous proteins in human cell models (Cao et al, 2023). This new design of crRNA could even suppress the development and metastasis of bladder cancer in mouse models by promoting the translation of anti-tumor proteins such as p53 and PTEN (Cao et al, 2023). In

another study, Eswarappa and colleagues demonstrated that dead Cas13 proteins from *Leptotrichia wadei* (LwaCas13a) and *Porphyromonas gulae* (PguCas13b) can induce stop codon readthrough across premature termination codons in several mRNAs (Manjunath et al, 2024). Recently, Iwasaki and colleagues showed the applicability of dCas13 to block translation in human cells for the first time (Apostolopoulos et al, 2024), but still, it is elusive whether the method can be generally used for various endogenous mRNAs. In our experimental condition, only U1-driven crRNAs but not U6-driven crRNAs were effective for the translational inhibition of EGFP (Fig. 5A), suggesting that cytosolic crRNAs are essential for the robustness of the gene-specific translational repression.

Similar strategies shall be applicable to Type III RNA-targeting Csm complexes. Since Type III systems are composed of multiple proteins rather than a single effector protein, they have rarely been repurposed in mammalian cells. Interestingly, in 2023, Doudna and colleagues demonstrated that a single plasmid containing all Csm components is effective in human cells for precise knockdown and live-cell RNA imaging (Colognori et al, 2023). The same group further achieved a single-molecule resolution of the live-cell RNA imaging using exceptionally long precursor crRNA arrays, which is a strong point of the Csm system (Xia et al, 2024). U1-driven crRNAs have the unique advantage of making a tiling pre-crRNA array because the spacer sequence can contain four consecutive U sequences, which is not favored in the U6 expression system.

The identification of diverse CRISPR-Cas systems has facilitated remarkable advancements in biology and medicine. Extensive endeavors in protein and RNA engineering have been dedicated to enhancing efficiency for human cell applications. The incorporation of the U1-driven crRNA offers a significant enhancement in efficiency that can be seamlessly integrated with existing engineered elements.

# Methods

**Reagents and tools table**

| Reagent/Resource | Reference or Source | Identifier or Catalog Number |
|---|---|---|
| **Experimental models** | | |
| HEK293E cell line | Jae-Sung Woo Lab | N/A |
| HeLa cell line | ATCC | CCL-2 |
| **Recombinant DNA** | | |
| All plasmids | Dataset EV1 | Dataset EV1 |
| **Antibodies** | | |
| Rabbit anti-HA | Cell Signaling | 3724 |
| Alexa Fluor-647 goat anti-rabbit | Invitrogen | A32733 |
| **Oligonucleotides and other sequence-based reagents** | | |
| Oligonucleotide list | Dataset EV1 | Dataset EV1 |
| Random hexamer | Invitrogen | 48190-011 |
| **Chemicals, Enzymes and other reagents** | | |

| Reagent/Resource | Reference or Source | Identifier or Catalog Number |
|---|---|---|
| Acid-Phenol:Chloroform:IAA (125:24:1) | Invitrogen | AM9720 |
| BbsI | Anza | IVGN0046 |
| DNase I | TaKaRa | 2270 |
| Dulbecco's Modified Eagle's Medium | Thermo Scientific | 12100061 |
| Fetal Bovine Serum | Gibco | 10270106 |
| GeneJuice Transfection Reagent | Sigma-Aldrich | 70967 |
| Glutathione magnetic agarose beads | Thermo Scientific | 78601 |
| Maxima H Minus Reverse Transcriptase | Thermo Scientific | EP0753 |
| PCRClean DX | Aline Biosciences | C-1003-250 |
| ProLong Diamond Antifade Mountant with DAPI | Invitrogen | P36966 |
| Quick-RNA Miniprep Kit | Zymo Research | R1055 |
| RevertAid reverse transcriptase | Thermo Scientific | EP0442 |
| SuperScript IV reverse transcriptase | Invitrogen | 18090200 |
| TB Green Premix Ex Taq II | TaKaRa | RR420 |
| TRIzol reagent | Invitrogen | 15596018 |
| T4 DNA Ligase | TaKaRa | 2011 |
| T4 RNA Ligase 2, truncated KQ | NEB | M0373 |
| UltraBrite Cypridina-Gaussia Dual Luciferase Assay Kit | Targeting Systems | DLAR-4 SG-1000 |
| **Software** | | |
| Fiji | https://imagej.net/software/fiji/ | Version v2.14 |
| FlowJo | https://www.flowjo.com | Version 10.8.1 |
| Prism | https://www.graphpad.com | Version 9.5.0 |
| cutadapt | https://anaconda.org/bioconda/cutadapt | Version 3.4 |
| fastx_toolkit | https://anaconda.org/bioconda/fastx_toolkit | Version 0.0.14 |
| pandas | https://pandas.pydata.org/ | Version 1.1.3 |
| snakemake | https://anaconda.org/bioconda/snakemake | Version 5.32.2 |
| python | https://www.python.org/ | Version 3.8.18 |
| matplotlib | https://matplotlib.org/ | Version 3.4.2 |
| **Other** | | |
| CFX Opus 384 | Bio-Rad | N/A |
| LSM900 inverted confocal microscope | ZEISS | N/A |
| NovoCyte Advanteon flow cytometer | Agilent | N/A |

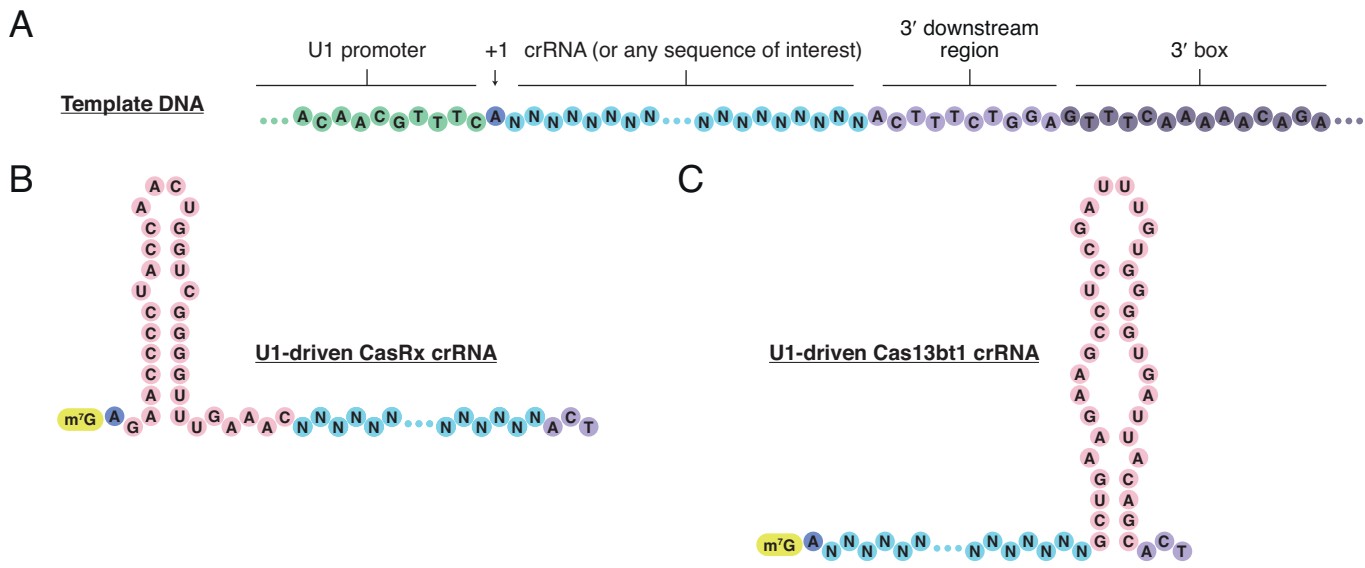

**Figure 6. Cytosolic Cas13 crRNAs expressed from U1 promoter.**

(A) Schematic of the U1 promoter and 3′ associated elements for crRNA. (B) Example of U1-driven CasRx crRNA. (C) Example of U1-driven Cas13bt1 crRNA.

## Plasmids

Human and mouse U1 promoter and associated 3′ elements were amplified from HAP1 and A3-1 genomic DNA, respectively. The CasRx direct repeat and BbsI cloning sites were added using overlap extension PCR. The amplicons were inserted into pUC19 using EcoRI and HindIII to make human and mouse U1 promoter-driven CasRx crRNA backbones (Fig. EV1). BbsI, EcoRI, and HindIII cloning sites on the human U1 promoter-driven CasRx crRNA backbone were used for the cloning of Cas13bt1, PspCas13b, and RanCas13b crRNA backbones. Individual crRNA spacers were cloned into the crRNA backbones using the BbsI cloning sites. Oligonucleotide sequences and plasmid maps are provided in Dataset EV1. crRNA backbone plasmids have been deposited at Addgene.

## Cell culture and transfection

HEK293E cells (human embryonic kidney 293 EBNA1; authenticated by ATCC STR profiling) and HeLa cells (ATCC #CCL-2) were cultured in Dulbecco's Modified Eagle's Medium (DMEM) (Thermo Scientific) supplemented with 10% Fetal Bovine Serum (FBS) (Gibco). All cultures were incubated at 37 °C in 5% $CO_2$ and 95% humidified air. According to the manufacturer's protocol, cells were transfected with plasmids using GeneJuice Transfection Reagent (Sigma-Aldrich) or the calcium phosphate transfection method. Cells were harvested 48 h after transfection for experiments.

## qRT-PCR

The Quick-RNA Miniprep Kit (Zymo) was used to extract total RNA according to the manufacturer's protocol. Reverse transcription was carried out using random hexamer (Invitrogen) and RevertAid reverse transcriptase (Thermo Scientific) according to the manufacturer's protocol. TB Green Premix Ex Taq II (TaKaRa)

was utilized for qPCR according to the manufacturer's protocol on CFX Opus 384 (Bio-Rad). β-Actin was used as the endogenous control. Oligonucleotide sequences are provided in Dataset EV1.

## Flow cytometry

Cells were plated and transfected with 200 ng plasmid DNA in 24-well plates. The transfection mix was composed of 40 ng of pEGFP-C1, 80 ng of plasmid for the corresponding Cas 13 protein, and 80 ng of plasmid for the corresponding crRNA. 48 h after transfection, cells were washed with PBS, trypsinized, and resuspended in the culture media. After centrifugation, the cell pellet was resuspended in the flow cytometry buffer (1× PBS, 2% FBS, 2 mM EDTA, and 2 mM $NaN_3$) and analyzed on NovoCyte Advanteon BVR flow cytometer (Agilent).

## Northern blot

TRIzol reagent (Invitrogen) was used to extract total RNA according to the manufacturer's protocol. For northern blotting of crRNAs, 30 μg of RNA was separated by 12.5% urea-polyacrylamide gel electrophoresis. Decade Marker (Invitrogen) was loaded as an RNA ladder. RNA was transferred to a Hybond N+ membrane (Amersham) by semi-dry transfer in 0.5× TBE buffer. RNA was UV cross-linked to the membrane for 15 s twice in an XL-1500 UV crosslinker (Spectro-UV). Hybridization was carried out by rotation at 37 °C for 1 h with a [32]P-labeled probe in PerfertHyb Plus hybridization buffer (Sigma-Aldrich). The signal was then visualized on Typhoon 9410 Variable Mode Imager (GE Healthcare). Oligonucleotide sequences are provided in Dataset EV1.

## Dual-luciferase reporter assay

Cells were plated and transfected with 200 ng plasmid DNA in 24-well plates. The transfection mix was composed of 8 ng of pC0037

reporter plasmid (a gift from Feng Zhang, Addgene plasmid #181934), 96 ng of plasmid for corresponding Cas 13 protein, and 96 ng of plasmid for corresponding crRNA. Dual-luciferase activity was measured with the UltraBrite Cypridina-Gaussia Dual Luciferase Assay Kit (Targeting Systems, cat#DLAR-4 SG-1000) according to the manufacturer's protocol.

## Immunocytochemistry

Hela cells were plated on cover glasses in 24-well plates and transfected with 200 ng plasmid. The transfection mix was composed of 100 ng of plasmid for the corresponding Cas13 protein and 100 ng of plasmid for the corresponding crRNA, or 200 ng of Cas13 protein plasmid for protein-only samples. 48 h after transfection, cells were fixed in 4% formaldehyde (Sigma-Aldrich) in PBS for 10 min and permeabilized in 70% ethanol for 4 h. For blocking, cells were incubated with 500 μL of 1% bovine serum albumin (BSA) (Rockland) with 0.1% Tween-20 (Santa Cruz) in PBS (PBS-T) for 30 min. Cells were incubated with the rabbit anti-HA primary antibody (Cell Signaling, cat#3724) (1:100 in 1% BSA, PBS-T) for 1 h, and then with the Alexa Fluor-647 goat anti-rabbit secondary antibody (Invitrogen, cat#A32733) (1:100 in 1% BSA, PBS-T) for 45 min. Cover glasses were mounted on a SuperFrost Plus adhesion slide (Epredia) with ProLong Diamond Antifade Mountant with DAPI (Invitrogen). Imaging was performed with the LSM900 inverted confocal microscope (ZEISS).

## RNA FISH

HeLa cells were plated on cover glasses in 24-well plates and transfected with 200 ng plasmid. The transfection mix was composed of 100 ng of plasmid for the corresponding Cas13 protein and 100 ng of plasmid for the corresponding crRNA, or 200 ng of crRNA plasmid for crRNA-only samples. 48 h after transfection, cells were fixed in 4% formaldehyde (Sigma-Aldrich) in PBS for 10 min and permeabilized in 70% ethanol for 4 h. Cover glasses were incubated in the ATTO-488 probe-containing hybridization buffer (2× SSC, 10% formamide, 10% dextran sulfate, 0.5 mg/mL yeast tRNA, 50 μg/mL BSA, and 10 mM ribonucleoside vanadyl complexes) overnight at 4 °C in a humidified chamber, and then mounted on a SuperFrost Plus adhesion slide (Epredia) with ProLong Diamond Antifade Mountant with DAPI (Invitrogen). Imaging was performed with the LSM900 inverted confocal microscope (ZEISS). For RNA FISH following immunocytochemistry, the fixation step was performed immediately after secondary antibody staining and then the same RNA FISH protocol was used. Signal intensity were quantified using ImageJ. All quantifications were conducted in a blinded and random manner.

## RACE

TRIzol reagent (Invitrogen) was used to extract total RNA according to the manufacturer's protocol. For 5′ RACE, the 5′ adapter sequence was added during the reverse-transcription reaction using SuperScript IV reverse transcriptase (Invitrogen) in the presence of a template-switching oligo (TSO) at 37 °C for 1 h. cDNA was then used for two rounds of PCR to enrich crRNA and add Illumina indexes. The PCR product was purified with PCRClean DX (Aline Biosciences). For 3′ RACE, a pre-

adenylated 3′ adapter was ligated to total RNA using T4 RNA Ligase 2, truncated KQ (NEB) at 25 °C for 4 h, and reverse transcription was carried out using SuperScript III reverse transcriptase (Invitrogen) in the absence of TSO at 50 °C for 1 h. Illumina indexes were added during PCR as above. The final purified PCR products were mixed in a pooled library and sequenced with Illumina NovaSeq (paired-end 150) at Genewiz. Oligonucleotide sequences are provided in Dataset EV1.

## RACE analysis

After removing 3′ adapter sequence from FASTQ files using cutadapt (ver 3.4; -a <adapter> --trimmed-only -m 20), high-quality reads were selected using fastq_quality_filter (ver 0.0.14; -q 30 -p 95). Four random nucleotides at the 5′ end of the 3′ RACE adapter were trimmed using fastx_trimmer, and the 5′ adapter sequence was removed using cutadapt (-g <adapter> --trimmed-only). The top 20 abundant sequences per sample were used to find end positions and modifications. Oligonucleotide sequences are provided in Dataset EV1, and the processed read counts are provided in Datasets EV2, EV3.

## Purification of the recombinant GST-eIF4E$_{K119A}$ protein

The coding sequence of GST-eIF4E$_{K119A}$ was amplified from pGEX-2T-GST-eIF4E$_{K119A}$ (Addgene #112818) and subcloned into a pET vector. BL21(DE3) expressing GST-eIF4E$_{K119A}$ was induced with 0.1 mM IPTG at OD 2 and incubated at 25 °C for 16 h. The cells were lysed by sonication in 20 mM Tris pH 7.5, 150 mM NaCl, 10% glycerol, 1 μg/mL RNase A, 3 μg/mL Staphylococcal aureus nuclease, 5 mM CaCl$_2$, 0.1 mM PMSF, 5 mM DTT, and 1% Triton X-100. The protein was enriched by a Pierce Glutathione Agarose resin (Thermo) and purified through a HiTrap Heparin HP column (Cytiva). Subsequently, the eluted protein was further purified through a Superdex 200 increase 10/300 column (Cytiva) in 20 mM Tris pH 7.5, 150 mM NaCl, 2 mM DTT.

## Capped RNA pull-down assay

TRIzol reagent (Invitrogen) was used to extract total RNA according to the manufacturer's protocol. Total RNA was purified with DNase I (TaKaRa) and Acid-Phenol:Chloroform:IAA (125:24:1) (Invitrogen) according to the manufacturer's protocols. Glutathione magnetic agarose beads (Thermo Scientific) and homemade GST-eIF4E$_{K119A}$ were pre-mixed in the cap binding buffer (10 mM Tris pH 7.5, 150 mM NaCl, 0.1% IGEPAL CA-630, 1 mM DTT) at 4 °C for 2 h on a rotator. Purified total RNA was added to the pre-mix and incubated at 4 °C for 2 h on a rotator. The Quick-RNA Miniprep Kit (Zymo) was used to extract RNA in the input, supernatant, and bound fraction according to the manufacturer's protocol. For qRT-PCR of crRNA, reverse transcription was carried out using specific primers and Maxima H Minus Reverse Transcriptase (Thermo Scientific) according to the manufacturer's protocol. Oligonucleotide sequences are provided in Dataset EV1.

## Disease-associated variation analysis

NCBI ClinVar data (variant_summary.txt.gz) and RefSeq mRNA sequence (GRCh38_latest_rna.gbff.gz) were downloaded on 11 Apr 2024 and 21 Mar 2023, respectively. Pathogenic single-nucleotide

variations that induce amino acid changes were selected for further analysis. G-to-A and T-to-C changes on mRNA were considered therapeutic candidates by REPAIR (A > I) and RESCUE (C > U), respectively. RefSeq mRNA sequences were used to find upstream and downstream nucleotide sequences of SNV. The processed data are provided in Dataset EV4.

## Data availability

The datasets and computer code produced in this study are available in the following databases: RACE data: Gene Expression Omnibus GSE275245. Source codes for RACE analysis: GitHub (https://github.com/jimkwon/u1promoter).

The source data of this paper are collected in the following database record: biostudies:S-SCDT-10_1038-S44319-025-00399-4.

## Peer review information

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

## Acknowledgements

We thank Jonathan Kong for his support and advice, Jeffery Lau and Joseph Ho for technical help, and Hinson Cheung, Ben Ong, Zi-Wei Ye, and Chi-Ping Chan for critical discussion. We also thank the staff of the Centre for PanorOmic

Sciences (CPOS) in the LKS Faculty of Medicine at the University of Hong Kong for their technical assistance. This work is supported by the Research Grants Council (GRF #17110223) and the University of Hong Kong (Start-up fund to SCK).

## Author contributions

**Ezra C K Cheng**: Conceptualization; Data curation; Formal analysis; Investigation; Visualization; Methodology; Writing—original draft; Writing—review and editing. **Joe K C Lam**: Resources; Methodology. **S Chul Kwon**: Conceptualization; Supervision; Funding acquisition; Investigation; Visualization; Methodology; Writing—original draft; Writing—review and editing.

Source data underlying figure panels in this paper may have individual authorship assigned. Where available, figure panel/source data authorship is listed in the following database record: biostudies:S-SCDT-10_1038-S44319-025-00399-4.

## Disclosure and competing interests statement

The authors declare no competing interests.

# Expanded View Figures

**A**   Human U1 promoter-containing CasRx crRNA expression vector (pKW82.phU1-Rx)

<u>EcoRI</u>
**GAATTC**ctaaggaccagcttctttgggagagaacagacgcaggggcgggagggaaaaagggagaggcagacgtcacttcc

\*                 \*
**C**cttggcg**G**ctctggcagcagattggtcggttgagtggcagaaaggcagacggggactgggcaaggcactgtcggtgaca

 \* <u>DSE (Distal Sequence Element)</u>
 \*
tcac**A**gacagggcgacttctatgtagatgaggcagcgcagaggctgctgcttcgccacttgctgcttcgccacgaagg**A**g

ttcccgtgccctgggagcgggttcaggaccgctgatcggaagtgagaatcccagctgtgtgtcagggctggaaagggctc

 <u>PSE (Proximal Sequence Element)</u> +1→ Transcription
 start site
gggagtgcgcggggcaagtgaccgtgtgtgtaaagagtgaggcgtatgaggctgtgtcggggcagaggcacaacgtttc**A**

 <u>CasRx direct repeat (31 nt)</u> BbsI BbsI 3' downstream region 3' box
gaacccctaccaactggtcggggtttgaaacgg**GTCTTC**ga**GAAGAC**ctactttctggagtttcaaaaacagactgtacg
 ↰ ↱
 crRNA cloning site <u>HindIII</u>
 # #
c**T**aagggtcatatctttt**C**ttgtattggtttgtgtcttggttggtgtcttag**AAGCTT**

 \* Variations found in AP023930.1
 # Variations found in AC277963.1
 PSE (Proximal Sequence Element): Reviewed in Biochim Biophys Acta 2008, 1779:295
 DSE (Distal Sequence Element): Element D (snRNA gene enhancer) found in J Biol Chem 1987, 262:1795

**B**   Mouse U1 promoter-containing CasRx crRNA expression vector (pKW83.pmU1-Rx)

<u>EcoRI</u>
**GAATTC**ctcgagctaaagactgtgcatccgactcctacatttatgaaagtaaatgcctattgttagaacaaaaaaggcta

cagaacaaaaaacaaagcgaaataccatctgctttaggttcagtggtattttcccgctgacaggaggcgggttttttggg

tacaggaaacgagtcactatggaggcggtactatgtagatgagaattcaggagcaaactgggaaaagcaactgcttccaa

atatttgtgattttacagtgtagttttggaaaaactcttagcctaccaattcttctaagtgttttaaaatgtgggagcc

agtacacatgaagttatagagtgtgttttaatgaggcttaaatatttaccgtaactatgaaatgctacgcatatcatgctgt

 Transcription <u>CasRx direct repeat (31 nt)</u> BbsI BbsI 3' downstream region
 start site +1→
tcaggctccgtggccacgcaactc**A**gaacccctaccaactggtcggggtttgaaacgg**GTCTTC**ga**GAAGAC**ctgtttac
 ↰ ↱
 crRNA cloning site

 <u>3' box</u>
ttggtttaaaaatagcttgcactagcgataccgcgaatatggttattaggtttgttaggcacagtcgtgtcttactata

 <u>HindIII</u>
ga**AAGCTT**

---

**Figure EV1.  U1 promoter sequences for cytosolic crRNA.**

(**A**) Map of the human U1-driven CasRx crRNA backbone. (**B**) Map of the mouse U1-driven CasRx crRNA backbone.

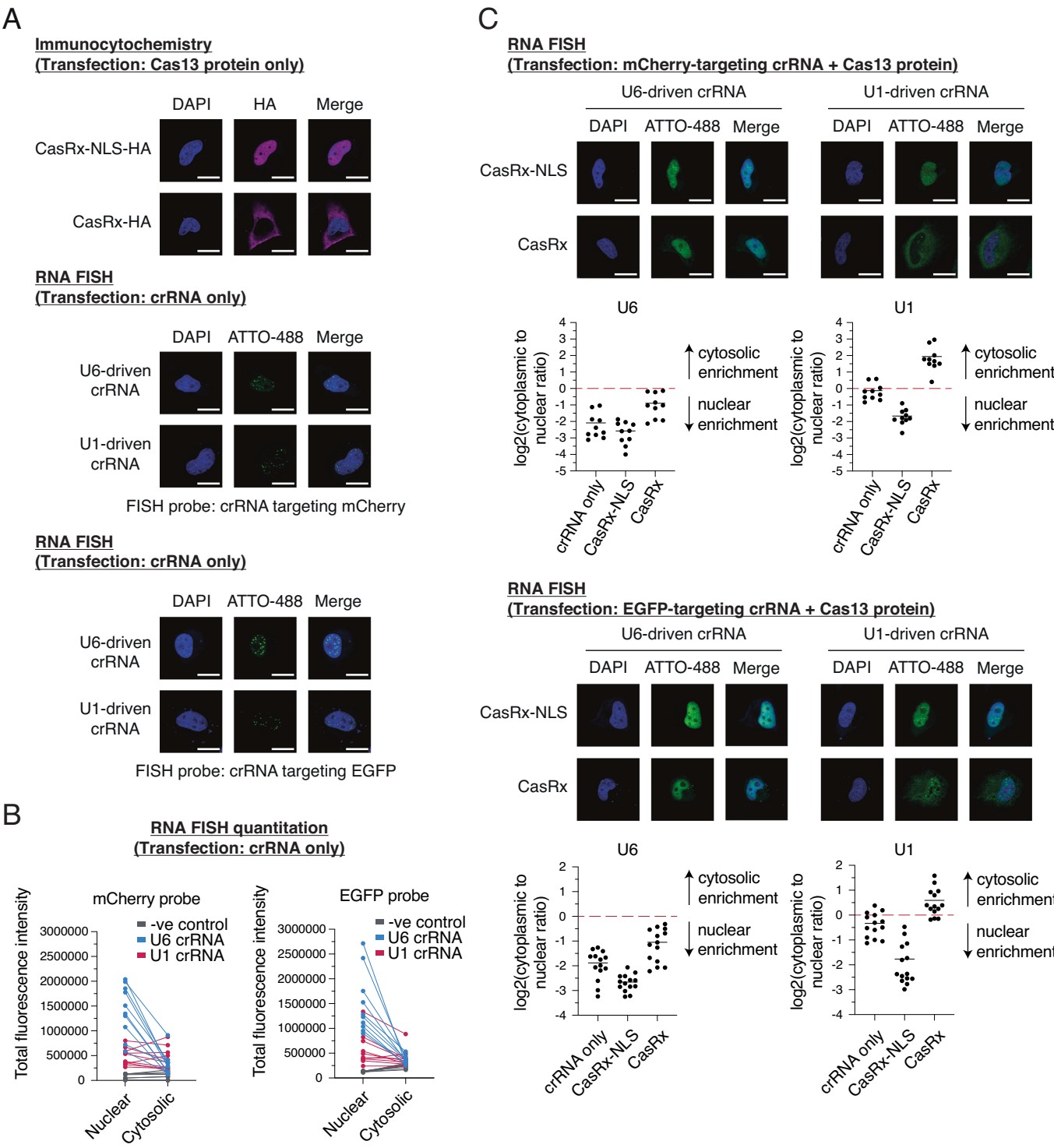

**Figure EV2. Repurposing U1 promoter for cytosolic crRNA.**

(**A**) Subcellular localization of crRNA and CasRx effector in the absence of the binding partner. Top, immunocytochemistry of CasRx-NLS and CasRx tagged with a HA tag. Middle and bottom, RNA FISH of crRNA. Nuclei were stained with DAPI. CasRx effector was stained with anti-HA antibody. crRNAs targeting mCherry and EGFP were visualized with ATTO-488-conjugated probes. Note that EGFP and mCherry were not expressed and acted as non-human sequences. Scale bars: 20 μm. (**B**) Quantitation of RNA FISH. crRNA signals in the absence of the CasRx protein are shown. $n = 10$ from two biological replicates. (**C**) Subcellular localization of crRNA in the presence of the CasRx protein. Top, crRNA targeting mCherry ($n = 10$ from two biological replicates). Bottom, crRNA targeting EGFP ($n = 14$ from two biological replicates). Note that mCherry and EGFP were not expressed and acted as non-human sequences. Total fluorescence intensity is shown as a scatter plot. Mean is indicated as a horizontal line. Scale bars: 20 μm. Source data are available online for this figure.

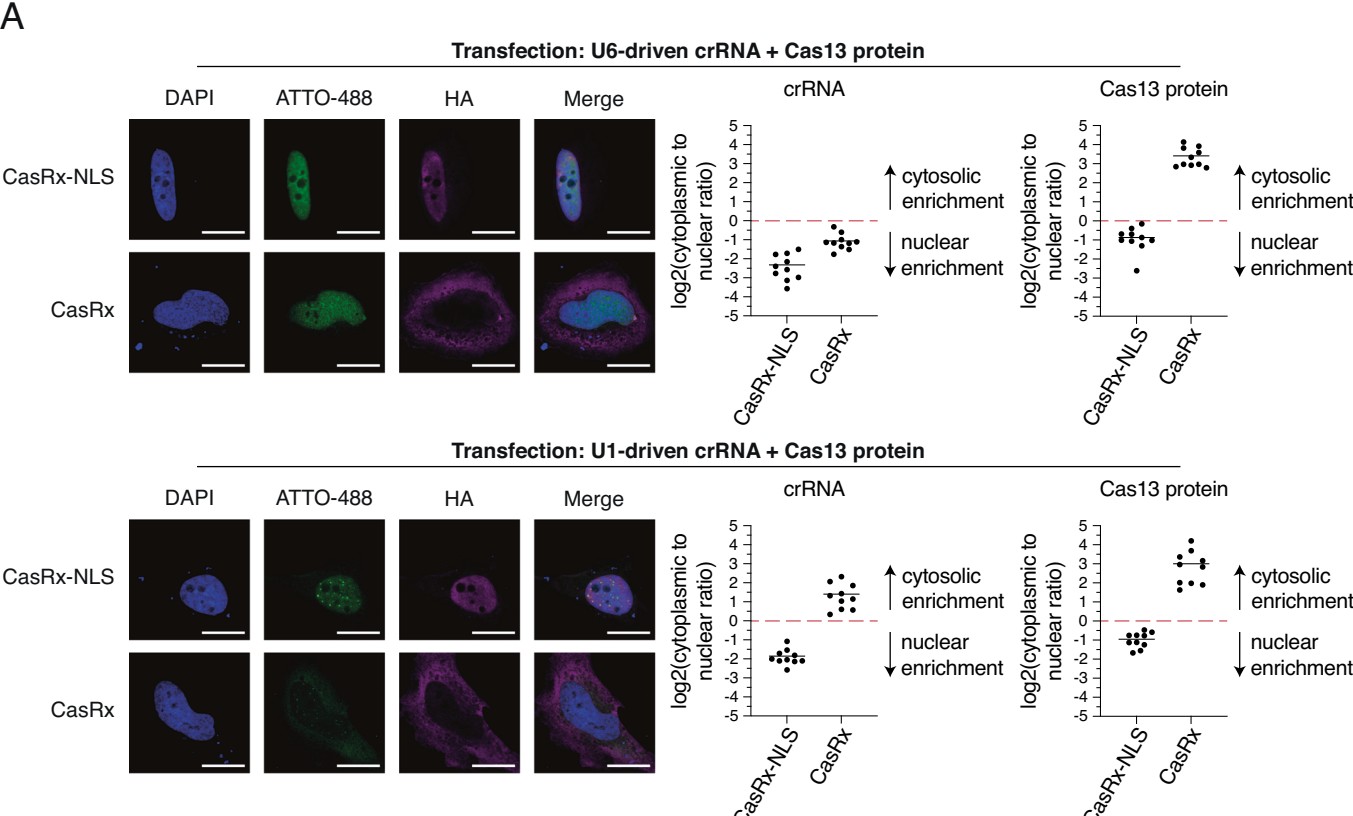

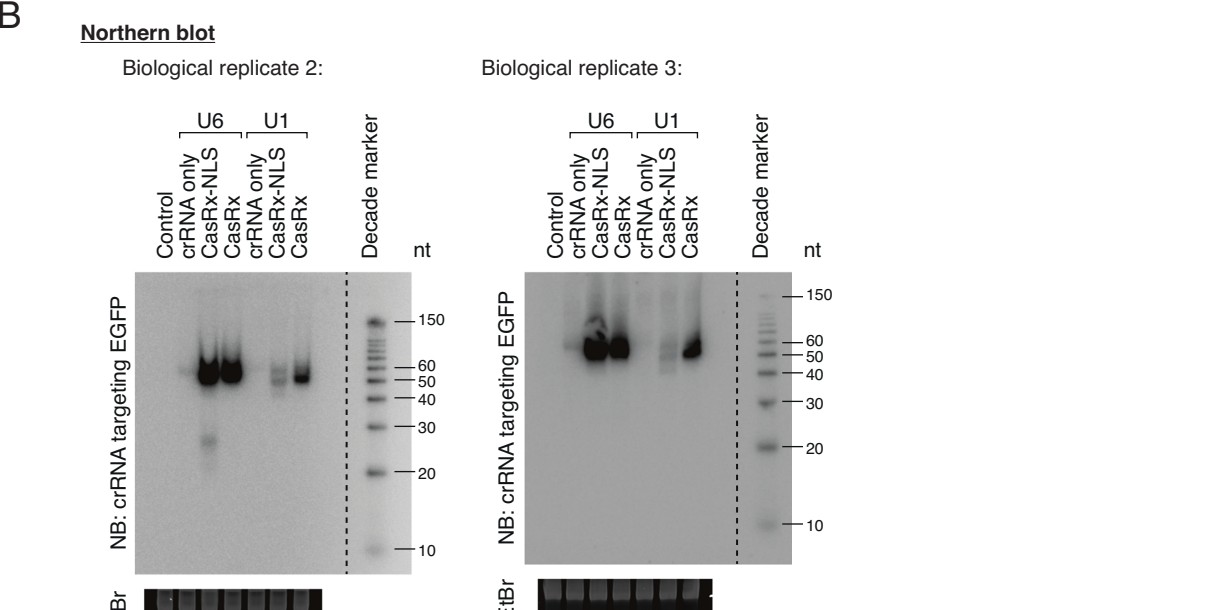

**Figure EV3. Subcellular localization of crRNA targeting EGFP.**

(A) Subcellular localization of crRNA targeting EGFP in the presence of the CasRx protein. RNA FISH was performed following immunocytochemistry of the same cells. Note that EGFP was not expressed and acted as a non-human sequence. Nuclei were stained with DAPI. crRNA targeting EGFP was visualized with an ATTO-488-conjugated probe. CasRx effector was stained with anti-HA antibody. Total fluorescence intensity is shown as a scatter plot. Mean is indicated as a horizontal line. $n = 10$ from three biological replicates. Scale bars: 20 μm. (B) Northern blot replicates showing crRNA expression and length distribution. crRNA targeting EGFP was visualized with a 32P-labeled probe. Note that EGFP was not expressed and acted as a non-human sequence. Source data are available online for this figure.

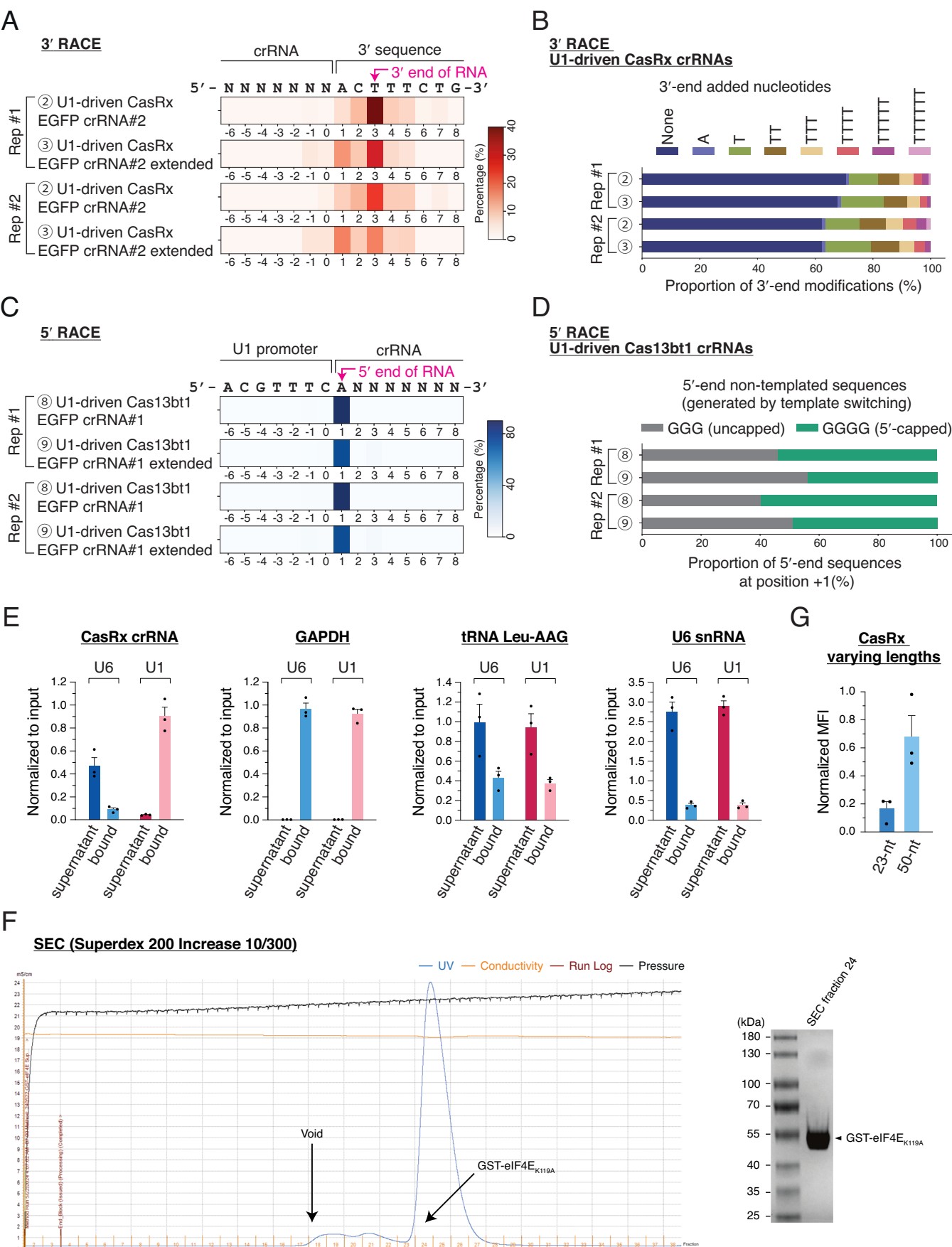

◀ **Figure EV4. Reproducibility of 5′ RACE and 3′ RACE experiments.**

(A) Analysis of 3′ end positions. The proportion of each 3′ end position is shown as a heatmap. (B) Analysis of 3′ end modifications. (C) Analysis of 5′ end positions. The proportion of each 5′ end position is shown as a heatmap. (D) Analysis of 5′ end non-templated sequences. (E) qRT-PCR following pull-down assay of capped RNA with recombinant mouse $eIF4E_{K119A}$ protein. Enrichment of U1-driven CasRx crRNA in the bound fraction is compared with U6-driven CasRx crRNA. GAPDH mRNA is included to represent capped RNA, while tRNA Leu-AGG and endogenous U6 snRNA are included to represent non-capped RNA. Mean and s.e.m. are shown ($n = 3$, biological triplicates). (F) Purification of GST-tagged $eIF4E_{K119A}$. Left, Size-exclusion chromatography (SEC). Right, SDS-PAGE showing homogeneity of the purified GST-tagged $eIF4E_{K119A}$. (G) EGFP knockdown by CasRx with varying U1-driven crRNA lengths. Median fluorescence intensity (MFI) is shown. Mean and s.e.m. are shown ($n = 3$, biological triplicates). Source data are available online for this figure.

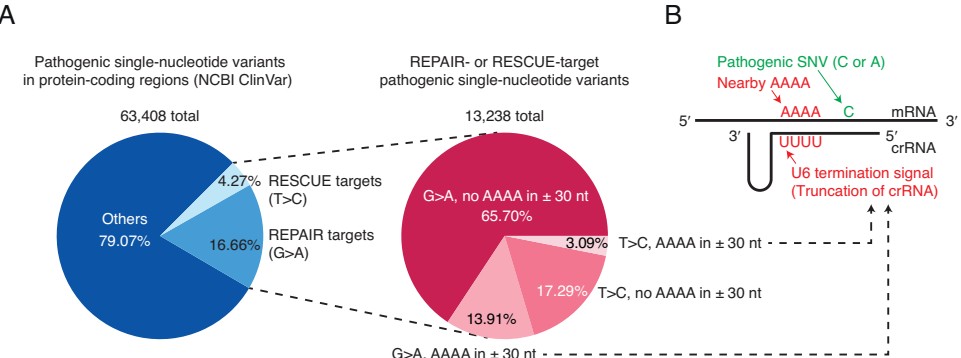

**Figure EV5. Targetable pathogenic SNVs (single-nucleotide variants) for REPAIR and RESCUE.**

(A) ClinVar database was screened for pathogenic SNV that could be targeted by the REPAIR and RESCUE RNA-editing systems. (B) Association of the targetable SNVs with adjacent AAAA sequences.

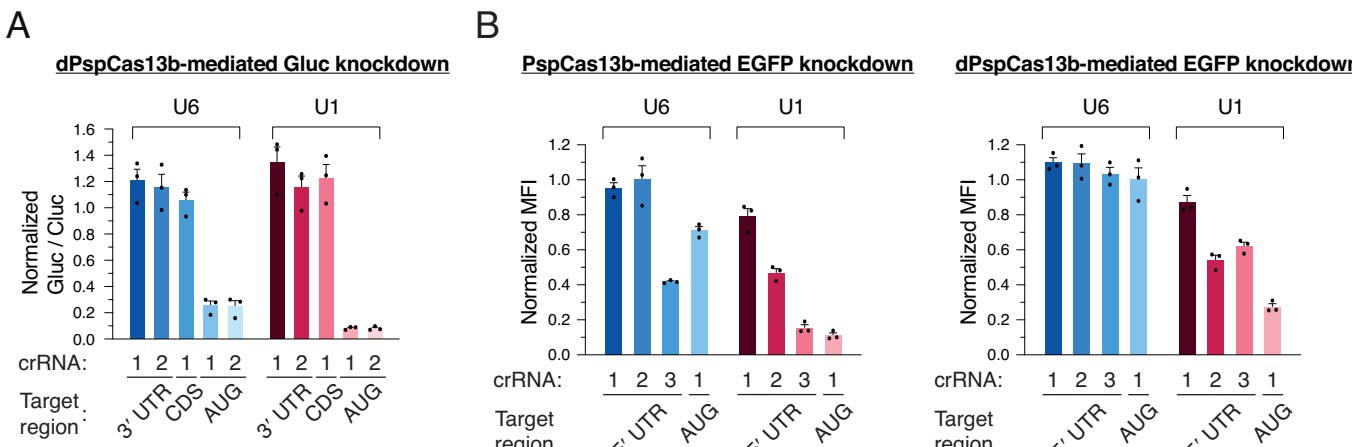

**Figure EV6. Enhancing translational repression using U1-driven crRNA.**

(A) Translational repression of *Gaussia* luciferase using AUG-targeting crRNA and catalytically dead PspCas13b. Mean and s.e.m. are shown ($n = 3$, biological triplicates).
(B) Translational repression of EGFP using 5′ UTR-targeting crRNA and catalytically dead PspCas13b. Repression efficiency with U1-driven crRNA is compared with the efficiency with U6-driven crRNA. Median fluorescence intensity (MFI) is normalized to the non-targeting control. Mean and s.e.m. are shown ($n = 3$, biological triplicates). Source data are available online for this figure.

