## [Peer Review File · EMBO Reports]

Cytosolic CRISPR RNAs for efficient application of RNA-targeting CRISPR-Cas systems

Ezra C.K. Cheng, Joe K.C. Lam, and S. Chul Kwon

Corresponding author(s): S. Chul Kwon (chul@hku.hk)

Review Timeline:

Submission Date:	16th Sep 24
Editorial Decision:	24th Oct 24
Revision Received:	20th Jan 25
Editorial Decision:	3rd Feb 25
Revision Received:	4th Feb 25
Accepted:	7th Feb 25

Editor: *Esther Schnapp*

Transaction Report:

Dear Prof. Kwon,

Thank you for the submission of your manuscript to EMBO reports. We have now received the full set of referee reports that is pasted below.

As you will see, the referees acknowledge that the findings are interesting. However, they also have several suggestions for how the study could be improved, and I think all suggestions are good and should be addressed. Please let me know in case you disagree and we can discuss the exact revision requirements further, also in a video chat, if you like.

I would thus like to invite you to revise your manuscript with the understanding that the referee concerns must be fully addressed and their suggestions taken on board. Please address all referee concerns in a complete point-by-point response. Acceptance of the manuscript will depend on a positive outcome of a second round of review. It is EMBO reports policy to allow a single round of major revision only and acceptance or rejection of the manuscript will therefore depend on the completeness of your responses included in the next, final version of the manuscript.

We realize that it is difficult to revise to a specific deadline. In the interest of protecting the conceptual advance provided by the work, we recommend a revision within 3 months (24th Jan 2025). Please discuss the revision progress ahead of this time with the editor if you require more time to complete the revisions.

- 1) A data availability section providing access to data deposited in public databases is missing. If you have not deposited any data, please add a sentence to the data availability section that explains that.
- 2) Your manuscript contains statistics and error bars based on $n=2$. Please use scatter blots in these cases. No statistics should be calculated if $n=2$.

5) a complete author checklist, which you can download from our author guidelines <https://www.embopress.org/page/journal/14693178/authorguide>. Please insert information in the checklist that is also reflected in the manuscript. The completed author checklist will also be part of the RPF.

6) Please note that all corresponding authors are required to supply an ORCID ID for their name upon submission of a revised manuscript (<https://orcid.org/>). Please find instructions on how to link your ORCID ID to your account in our manuscript tracking system in our Author guidelines <https://www.embopress.org/page/journal/14693178/authorguide#authorshipguidelines>

7) Before submitting your revision, primary datasets produced in this study need to be deposited in an appropriate public

database (see <https://www.embopress.org/page/journal/14693178/authorguide#datadeposition>). Please remember to provide a reviewer password if the datasets are not yet public. The accession numbers and database should be listed in a formal "Data Availability" section placed after Materials & Method (see also <https://www.embopress.org/page/journal/14693178/authorguide#datadeposition>). Please note that the Data Availability Section is restricted to new primary data that are part of this study. * Note - All links should resolve to a page where the data can be accessed. *

10) Regarding data quantification (see Figure Legends:

<https://www.embopress.org/page/journal/14693178/authorguide#figureformat>)

12) All Materials and Methods need to be described in the main text using our 'Structured Methods' format, which is required for all research articles. According to this format, the Methods section includes a Reagents and Tools Table (listing key reagents, experimental models, software and relevant equipment and including their sources and relevant identifiers) followed by a Methods and Protocols section describing the methods using a step-by-step protocol format. The aim is to facilitate adoption of the methodologies across labs. More information on how to adhere to this format as well as a downloadable template (.docx) for the Reagents and Tools Table can be found in our author guidelines:

An example of a Method paper with Structured Methods can be found here: <https://www.embopress.org/doi/full/10.1038/s44320-024-00037-6#sec-4>

You are able to opt out of this by letting the editorial office know (emboreports@embo.org). If you do opt out, the Review

Process File link will point to the following statement: "No Review Process File is available with this article, as the authors have chosen not to make the review process public in this case."

I look forward to seeing a revised form of your manuscript when it is ready.

Yours sincerely,

Referee #1:

* General comments;

RNA targeting or editing via CRISPR-Cas13 are promising for gene regulation as well as disease treatment. In this study, Ezra et al. utilized a U1 snRNA promoter system for guide RNA expression, which is mediated by RNA polymerase II, instead of a typical U6 promoter system mediated by RNA polymerase III. Owing to the unique features of the U1 snRNA promoter, expressed guide RNAs are abundant in the cytoplasm rather than the nucleus, making it better for RNA targeting or RNA editing of interest. Overall, the topic of this manuscript is quite interesting for board researchers, and the manuscript is straightforward and well written. The manuscript appears to be suitable for this journal, EMBO Reports, but several issues need to be addressed prior to publication.

* Specific comments;

1) As the authors mentioned in the introduction section, there are several types of RNA polymerase II-dependent promoters such as CMV, EF1 α , and U1 promoters. But the authors only selected and tested the U1 promoter only. What are merits of U1 promoters compared to other RNA polymerase II-dependent promoters such as CMV and EF1 α ? If necessary, comparison experiments would be required.

2) In Figure 1, CasRx with NLS or NES exhibited better activity than CasRx-NES. Considering that the guide RNA expression under the U1 snRNA promoter has benefits of cytosolic export, reasons or discussion to explain the results are required here.

3) I am curious how the RNA polymerase II regulate the termination of the transcription process. To my understanding, it seems that 3 nucleotides (i.e., "ACT") have a relation with the transcription termination process of the U1 promoter shown in Figure 2. Doesn't the addition of these sequences affect the CRISPR-Cas13 structure or editing efficiency? It should be discussed.

4) Typos in Figure 4C: RNA-edting -> RNA-editing

Referee #2:

Due to the specific target RNA recognition, the CRISPR-Cas13 system offers promising molecular tools to regulate mRNAs without altering genomic DNA. Despite the diverse applications developed over the past several years, further enhancement of efficacy is a demanding task. In this manuscript, the authors showed that a U1 promoter, rather than the commonly used U6 promoter, provides more potent effects of Cas13-based tools since U1-driven gRNAs localize in cytoplasm more effectively. The paper is well-written and thoughtfully presented. This reviewer supports this manuscript for publication but has several minor comments listed below.

1.

It is clearly demonstrated that the U1 promoter alters the localization of gRNA to cytoplasm and gRNA is stabilized by co-transfected Cas13. However, the co-localization of gRNA with Cas13 or dCas13 was not validated. This presentation would further support the manuscript's claims and strengthen the conclusion.

2.

The authors interpreted that 4Gs tracts added during template switching originated from m7G cap. Although this was totally reasonable, it would be helpful if the author could assess the presence of m7G cap on the gRNAs. CAGE or equivalent techniques could assess this. Ideally, the fraction of capped gRNA would be worth reporting.

Referee #3:

This is an excellent paper. The authors have found that the use of a U1 promoter allows for Cas13 crRNAs to be efficiently

exported to the cytoplasm, rather than remaining in the nucleus. This is very helpful for applications where one wants to target cytoplasmic RNAs. The authors demonstrate RNA knockdown and editing, but I am sure this will be useful to others in the field.

The paper is clearly written, and the experiments are quite logical overall. My only suggestion might be to move some of the RACE experiments/analysis to the supplement. While these results are quite interesting, I found the other data to be even more compelling.

Authors' Response to Reviewers

We thank the reviewers for their thoughtful reviews and comments. Please see the point-by-point responses below. The original comments are highlighted in blue.

Referee #1:

* General comments;

RNA targeting or editing via CRISPR-Cas13 are promising for gene regulation as well as disease treatment. In this study, Ezra et al. utilized a U1 snRNA promoter system for guide RNA expression, which is mediated by RNA polymerase II, instead of a typical U6 promoter system mediated by RNA polymerase III. Owing to the unique features of the U1 snRNA promoter, expressed guide RNAs are abundant in the cytoplasm rather than the nucleus, making it better for RNA targeting or RNA editing of interest. Overall, the topic of this manuscript is quite interesting for board researchers, and the manuscript is straightforward and well written. The manuscript appears to be suitable for this journal, EMBO Reports, but several issues need to be addressed prior to publication.

* Specific comments;

1) As the authors mentioned in the introduction section, there are several types of RNA polymerase II-dependent promoters such as CMV, EF1 α , and U1 promoters. But the authors only selected and tested the U1 promoter only. What are merits of U1 promoters compared to other RNA polymerase II-dependent promoters such as CMV and EF1 α ? If necessary, comparison experiments would be required.

Response:

We thank Referee #1 for pointing out the necessity to distinguish the U1 promoter from the other common RNAPII-dependent promoters. A key and fundamental difference is that the other promoters, such as CMV and EF1 α , rely on the polyadenylation signal and associated pathway for the 3' end processing of mRNAs. However, the addition of poly(A) tail is neither ideal nor necessary for efficient expression of short crRNA. Therefore, the U1 promoter is preferred here because an alternative 3' end processing pathway is used and the homogeneity of resulting 3' ends is demonstrated in Fig. 2B. We have added relevant discussion in the introduction of the revised manuscript as below.

Line 54: In contrast to the polyadenylation signal-dependent 3' end processing of CMV or EF1 α RNAPII-transcripts, the 3' box-dependent processing pathway utilizes a distinct cleavage complex called Integrator, enabling efficient expression of short RNA species without a long poly(A) tail (Chen & Wagner, 2010; Matera & Wang, 2014).

Line 85: The canonical mRNA promoters, such as CMV and EF1 α promoters, also have been shown to be useful for multiplexed crRNA expression for Cas12a, which contains the pre-crRNA processing activity; however, it is still controversial whether these RNAPII promoters outperform the U6 promoter for single gene regulation. (Campa et al, 2019; Zhao et al, 2024; Zhong et al, 2017).

2) In Figure 1, CasRx with NLS or NES exhibited better activity than CasRx-NES. Considering that the guide RNA expression under the U1 snRNA promoter has benefits of cytosolic export, reasons or discussion to explain the results are required here.

Response:

Some N-terminal or C-terminal tags affect protein function. We speculate that the hydrophobic NES residues (LYPERLRRILT) may interfere with the proper folding of the protein or the binding of RNA. We showed that CasRx without any localization signal mainly localizes in the cytoplasm. Therefore, we believe that both CasRx and CasRx-NES localize in the cytoplasm, but CasRx-NES may have a protein folding problem or an RNA binding issue compared to a natural CasRx protein. Since there are other NES sequences, it would be interesting to test various NES-CasRx fusion proteins to improve the activity.

We added this clause at the end of the previous sentence:

Line 165: *We also included a C-terminal NES (nuclear export signal)-fused CasRx construct and obtained the same pattern, albeit its overall knockdown efficiency was lower than that of CasRx without NLS, suggesting that the hydrophobic NES residues may affect the protein folding or the RNA binding (Fig. 1F).*

3) I am curious how the RNA polymerase II regulate the termination of the transcription process. To my understanding, it seems that 3 nucleotides (i.e., "ACT") have a relation with the transcription termination process of the U1 promoter shown in Figure 2. Doesn't the addition of these sequences affect the CRISPR-Cas13 structure or editing efficiency? It should be discussed.

Response:

Thank you for raising this issue for broad readers. We included a paragraph in Discussion to explain this important issue.

Line 359: *The 3' RACE analysis of U1-driven crRNA in the absence of the Cas13 effector protein shows the existence of the extra "ACT" sequences derived from the 3' downstream region of the U1 promoter at the 3' end of crRNA (Figs. 2B and 6A). However, the addition of these extra sequences does not seem to affect the function of crRNA in cells. For the Cas13b family members, the pre-crRNA processing activity of the Cas13b protein will cleave the crRNA after direct repeat, releasing the extra downstream nucleotides (Slaymaker et al, 2019). For U1-driven CasRx crRNA, the 3' extra sequences seem to be trimmed in the cytosol by exonucleases, as shown in the northern blot (Fig. 1D, lanes 6 and 7).*

4) Typos in Figure 4C: RNA-edting -> RNA-editing

Response:

We thank Referee #1 for the careful reading.

Referee #2:

Due to the specific target RNA recognition, the CRISPR-Cas13 system offers promising molecular tools to regulate mRNAs without altering genomic DNA. Despite the diverse applications developed over the past several years, further enhancement of efficacy is a demanding task. In this manuscript, the authors showed that a U1 promoter, rather than the commonly used U6 promoter, provides more potent effects of Cas13-based tools since U1-driven gRNAs localize in cytoplasm more effectively. The paper is well-written and thoughtfully presented. This reviewer supports this manuscript for publication but has several minor comments listed below.

1.

It is clearly demonstrated that the U1 promoter alters the localization of gRNA to cytoplasm and gRNA is stabilized by co-transfected Cas13. However, the co-localization of gRNA with Cas13 or dCas13 was not validated. This presentation would further support the manuscript's claims and strengthen the conclusion.

Response:

We thank Referee #2 for the positive feedback. We agree with the importance of validating the co-localization of crRNA and Cas13 protein. In addition to our previous RNA FISH data, we have performed immunocytochemistry of the Cas13 protein and RNA FISH of the crRNA on the same sample to provide direct evidence of their co-localization. The new representative images and signal quantification are added as Fig. 1C (crRNA targeting the mCherry sequence) and Appendix Fig. S1A (crRNA targeting the EGFP sequence).

Fig. 1C. Subcellular localization of crRNA targeting mCherry in the presence of the CasRx protein. RNA FISH was performed following immunocytochemistry of the same cells. Note that mCherry was not expressed and acted as a non-human sequence. Nuclei were stained with DAPI. crRNA targeting mCherry was visualized with an ATTO-488-conjugated probe. CasRx effector was stained with anti-HA antibody. Total fluorescence intensity is shown as a scatter plot. Mean is indicated as a horizontal line. $n = 10$ from three biological replicates. Scale bars: 20 μm .

Appendix

A

Appendix Fig. S1A. Subcellular localization of crRNA targeting EGFP in the presence of the CasRx protein. RNA FISH was performed following immunocytochemistry of the same cells. Note that EGFP was not expressed and acted as a non-human sequence. Nuclei were stained with DAPI. crRNA targeting EGFP was visualized with an ATTO-488-conjugated probe. CasRx effector was stained with anti-HA antibody. Total fluorescence intensity is shown as a scatter plot. Mean is indicated as a horizontal line. $n = 10$ from three biological replicates. Scale bars: 20 μm .

In the new Fig. 1C and Appendix Fig. S1A, the same localization pattern of crRNA in the presence of CasRx protein is observed as in the previous RNA FISH-only data (now moved to Fig. EV2C). With the simultaneous staining of Cas13 protein this time, we can validate the co-localization of U1-driven crRNA and the CasRx protein. We appreciate Referee #2's suggestion to strengthen the validation.

2.

The authors interpreted that 4Gs tracts added during template switching originated from m⁷G cap. Although this was totally reasonable, it would be helpful if the author could assess the presence of m⁷G cap on the gRNAs. CAGE or equivalent techniques could assess this. Ideally, the fraction of capped gRNA would be worth reporting.

Response:

To demonstrate the presence of an m⁷G cap on U1-driven crRNA, we have carried out an *in vitro* pull-down assay of capped RNA species using the recombinant GST-eIF4E_{K119A} protein. The relative portions of RNA in the supernatant and bound fractions are compared by RNA extraction and subsequent qRT-PCR in Fig. 3D (Cas13bt1 crRNA) and Fig. EV3E (CasRx crRNA). Fig. EV3F shows the purification of the recombinant GST-eIF4E_{K119A} protein.

Fig. 3D. qRT-PCR following pull-down assay of capped RNA with recombinant mouse eIF4E_{K119A} protein. Enrichment of U1-driven Cas13bt1 crRNA in the bound fraction is compared with U6-driven Cas13bt1 crRNA. GAPDH mRNA is included to represent capped RNA, while tRNA Leu-AGG and endogenous U6 snRNA are included to represent non-capped RNA. Mean and s.e.m. are shown ($n = 3$, biological triplicates).

Fig. EV3E. qRT-PCR following pull-down assay of capped RNA with recombinant mouse eIF4E_{K119A} protein. Enrichment of U1-driven CasRx crRNA in the bound fraction is compared with U6-driven CasRx crRNA. GAPDH mRNA is included to represent capped RNA, while tRNA Leu-AGG and endogenous U6 snRNA are included to represent non-capped RNA. Mean and s.e.m. are shown ($n = 3$, biological triplicates).

F

Fig. EV3F. Purification of GST-tagged eIF4E_{K119A}. Left, Size-exclusion chromatography (SEC). Right, SDS-PAGE showing homogeneity of the purified GST-tagged eIF4E_{K119A}.

In the main text, we added a paragraph as below.

Line 223: To validate the presence of m^7G cap on U1-driven crRNA, we performed an *in vitro* pull-down assay to capture 5' capped crRNA using the recombinant mouse eIF4E_{K119A} protein (Trotman et al, 2017) (Figs. 3D and EV3E,F). The K119A mutation has been shown to increase the cap binding affinity (Choi & Hagedorn, 2003). U1-driven Cas13bt1 crRNA showed a clear enrichment in the bound fraction as the m^7G capped GAPDH mRNA, while the pattern for U6-driven crRNA resembled the non-capped tRNA Leu-AAG and endogenous U6 snRNA transcripts (Fig. 3D). Similar enrichment pattern of U1-driven crRNA was reproduced using CasRx crRNA, demonstrating that the 5' capping of U1-driven crRNA is not effected by the 5' direct repeat hairpin structure (Fig. EV3E).

Although the new data does not provide a quantitative analysis of the proportion of capped crRNA, the qualitative comparison with controls strongly indicates that 5' capping is prevalent exclusively in U1-driven crRNA. We thank Referee #2 for allowing us to make a clear conclusion about the capping status of U1-driven crRNA.

Referee #3:

This is an excellent paper. The authors have found that the use of a U1 promoter allows for Cas13 crRNAs to be efficiently exported to the cytoplasm, rather than remaining in the nucleus. This is very helpful for applications where one wants to target cytoplasmic RNAs. The authors demonstrate RNA knockdown and editing, but I am sure this will be useful to others in the field.

The paper is clearly written, and the experiments are quite logical overall. My only suggestion might be to move some of the RACE experiments/analysis to the supplement. While these results are quite interesting, I found the other data to be even more compelling.

Response:

We thank Referee #3 for the positive comment and for the suggestion about emphasizing the other findings rather than the RACE analysis. However, we think it is better to keep the original Figs. 2 and 3 to show the reliability and consistency of the U1 promoter for diverse crRNA sequences and structures. If we show only a few examples in Figs. 2 and 3, readers may be concerned about whether they can use the U1 promoter to precisely express their own crRNAs. We appreciate Referee #3's suggestion.

Dear Prof. Kwon,

Thank you for the submission of your revised manuscript. We have now received the enclosed comments from referee 2 who was asked to assess your full response, and I am happy to say that this referee supports the publication of your study now. Only a few editorial requests will need to be addressed before we can proceed with the official acceptance of your manuscript:

- Please add up to 5 keywords to your ms file.
- Please move the Data Availability Section to before the Acknowledgments.
- Please correct the conflict of interest subheading to "Disclosure and Competing Interests Statement"
- Please remove the author credits from the ms file. All credits are entered during online ms submission.
- In the author checklist please answer all questions in the statistics section and send us a new, completed checklist.
- The 4 EV tables need to be called Datasets EV1-EV4; their legends should be removed from the ms file and should be provided in their respective Excel files as a separate sheet or tab.
- The APPENDIX Figure can be changed into Figure EV6, as we can offer more than 5 EV figures now. The legend needs to be removed from the Figure file and provided in the ms file (callouts and source file name and title will have to be updated too).

I would like to suggest some minor changes to the title and abstract. Please let me know whether you agree with the following:

Cytosolic CRISPR RNAs for efficient application of RNA-targeting CRISPR-Cas systems

Clustered regularly interspaced short palindromic repeats/CRISPR-associated protein (CRISPR/Cas) technologies have evolved rapidly over the past decade with the continuous discovery of new Cas systems. In particular, RNA-targeting CRISPR-Cas13 proteins are promising single-effector systems to regulate target mRNAs without altering genomic DNA, yet the current Cas13 systems are restrained by suboptimal efficiencies. Here, we show that U1 promoter-driven CRISPR RNAs (crRNAs) increase the efficiency of various applications, including RNA knockdown and editing, without modifying the Cas13 protein effector. We confirm that U1-driven crRNAs are exported into the cytoplasm, while conventional U6 promoter-driven crRNAs are mostly confined to the nucleus. Furthermore, we reveal that the end positions of crRNAs expressed by the U1 promoter are consistent regardless of guide sequences and lengths. We also demonstrate that U1-driven crRNAs, but not U6-driven crRNAs, can efficiently repress the translation of target genes in combination with catalytically inactive Cas13 proteins. Finally, we show that U1-driven crRNAs can counteract the inhibitory effect of miRNAs. Our simple and effective engineering enables unprecedented cytosolic RNA-targeting applications.

Referee #2:

This reviewer appreciates the authors' effort to improve the manuscript. The authors addressed all points raised in the previous review. This reviewer supports the manuscript for publication.

All editorial and formatting issues were resolved by the authors.

Prof. S. Chul Kwon
University of Hong Kong
Laboratory Block
21 Sassoon Road
Pokfulam, Hong Kong 00000
Hong Kong Special Administrative Region of the People's Republic of China

Dear Prof. Kwon,

I am very pleased to accept your manuscript for publication in the next available issue of EMBO reports. Thank you for your contribution to our journal.
